# Unique morphogenetic signatures define mammalian neck muscles and associated connective tissues

Eglantine Heude[1,2], Marketa Tesarova[3], Elizabeth M Sefton[4], Estelle Jullian[5], Noritaka Adachi[5], Alexandre Grimaldi[1,2], Tomas Zikmund[3], Jozef Kaiser[3], Gabrielle Kardon[4], Robert G Kelly[5], Shahragim Tajbakhsh[1,2]*

[1]Department of Developmental and Stem Cell Biology, Institut Pasteur, Paris, France; [2]CNRS UMR 3738, Paris, France; [3]Central European Institute of Technology, Brno University of Technology, Brno, Czech Republic; [4]Department of Human Genetics, University of Utah, Salt Lake City, United States; [5]Aix-Marseille Université, CNRS UMR 7288, IBDM, Marseille, France

**Abstract** In vertebrates, head and trunk muscles develop from different mesodermal populations and are regulated by distinct genetic networks. Neck muscles at the head-trunk interface remain poorly defined due to their complex morphogenesis and dual mesodermal origins. Here, we use genetically modified mice to establish a 3D model that integrates regulatory genes, cell populations and morphogenetic events that define this transition zone. We show that the evolutionary conserved cucullaris-derived muscles originate from posterior cardiopharyngeal mesoderm, not lateral plate mesoderm, and we define new boundaries for neural crest and mesodermal contributions to neck connective tissue. Furthermore, lineage studies and functional analysis of *Tbx1*- and *Pax3*-null mice reveal a unique developmental program for somitic neck muscles that is distinct from that of somitic trunk muscles. Our findings unveil the embryological and developmental requirements underlying tetrapod neck myogenesis and provide a blueprint to investigate how muscle subsets are selectively affected in some human myopathies.

DOI: https://doi.org/10.7554/eLife.40179.001

**Competing interests:** The authors declare that no competing interests exist.

## Introduction

The neck is composed of approximately 80 skeletal muscles in humans that allow head mobility, respiration, swallowing and vocalization processes, containing essential elements such as the trachea, esophagus, larynx, and cervical vertebrae. These processes are ensured by a robust network of muscles at the head-trunk interface, a transition zone subjected to a spectrum of human muscle diseases such as dropped head syndrome, oculopharyngeal myopathy, myotonic dystrophy, Duchenne-type dystrophy and congenital muscular disorders (*Emery, 2002*; *Martin et al., 2011*; *Randolph and Pavlath, 2015*). Defining the embryology of these distinct muscle groups is critical to understand the mechanisms underlying the susceptibility of specific muscles to muscular dystrophies. While myogenesis at the cranial and trunk levels has been studied extensively, the developmental mechanisms at the basis of neck muscle formation are poorly documented and often debated (*Ericsson et al., 2013*).

In vertebrates, head and trunk muscles arise from different mesodermal origins and their development depends on distinct myogenic programs. At the cranial level, the cardiopharyngeal mesoderm (CPM) resides in pharyngeal arches and gives rise to branchiomeric muscles and the second heart field. CPM specification is initiated by the activation of genes such as *Mesp1*, *Islet1* and *Tbx1*, while *Pax7* subsequently marks muscle stem cells (*Diogo et al., 2015*; *Kelly et al., 2004*; *Nathan et al.,*

2008; *Saga et al., 1996*; *Sambasivan et al., 2009*). In contrast, *Pax3* and *Pax7* are expressed in somitic mesoderm giving rise to trunk and limb muscles, with *Pax3* then being downregulated in most muscles during fetal stages, while *Pax7* maintains the stem cell pool (*Kassar-Duchossoy et al., 2005*; *Relaix et al., 2005*; *Tajbakhsh et al., 1997*). After the differential specification of cranial and trunk progenitors, the bHLH myogenic regulatory factors (MRFs) Myf5, Mrf4, Myod and Myog regulate myogenic cell fate and differentiation (reviewed in [*Comai and Tajbakhsh, 2014*; *Noden and Francis-West, 2006*]).

In early embryos, *Tbx1* is required for robust activation of MRF genes and proper branchiomeric muscle formation (*Grifone et al., 2008*; *Kelly et al., 2004*; *Kong et al., 2014*; *Sambasivan et al., 2009*). In *Tbx1* mutant embryos, the first pharyngeal arch is hypoplastic and posterior pharyngeal arches do not form resulting in variably penetrant defects of masticatory muscles and absence of muscles derived from more posterior arches (*Kelly et al., 2004*; *Lescroart et al., 2015*; *Theis et al., 2010*). In humans, *TBX1* is a major gene involved in 22q11.2 deletion syndrome (DiGeorge/velo-cardio-facial syndrome), a congenital disease characterized by cardiovascular defects and craniofacial malformations (*Papangeli and Scambler, 2013*). In contrast, *Pax3* acts upstream of MRF genes in somites and *Pax3* mutants have defects of epaxial and hypaxial muscle formation while double *Pax3/Pax7*-null embryos lack trunk/limb muscles (*Brown et al., 2005*; *Relaix et al., 2005*; *Tajbakhsh et al., 1997*; *Tremblay et al., 1998*).

The neck constitutes a transition zone characterizing land vertebrates (tetrapods). The major muscle groups in the neck consist of: epaxial back muscles; ventral hypaxial musculature; pharyngeal, laryngeal and esophagus striated muscles located medioventrally; and cucullaris-derived muscles. The cucullaris is a generic term defining putative homologous muscles that are evolutionarily conserved and connect the head and trunk in jawed vertebrates (gnathostomes). In amniotes, the cucullaris represents the embryonic anlage that gives rise to trapezius and sternocleidomastoid muscles which are innervated by the accessory nerve XI (*Diogo, 2010*; *Edgeworth, 1935*; *Ericsson et al., 2013*; *Kuratani, 2008*; *Kuratani et al., 2018*; *Lubosch, 1938*; *Tada and Kuratani, 2015*).

While the somitic origin of epaxial/hypaxial neck muscles and CPM origin of pharyngeal, laryngeal and esophagus striated muscles are well defined (*Gopalakrishnan et al., 2015*; *Noden, 1983*; *Tabler et al., 2017*), the embryological origin of cucullaris-derived muscles has remained a subject of controversy (*Couly et al., 1993*; *Edgeworth, 1935*; *Greil, 1913*; *Huang et al., 1997*; *Huang et al., 2000*; *Matsuoka et al., 2005*; *Noden, 1983*; *Piatt, 1938*; *Piekarski and Olsson, 2007*). This muscle group was reported to originate either from lateral plate mesoderm (LPM) or CPM populations adjacent to the first three somites in chick and axolotl (*Nagashima et al., 2016*; *Sefton et al., 2016*; *Theis et al., 2010*). However, retrospective lineage analysis indicated that the murine trapezius and sternocleidomastoid muscles are clonally related to second heart-field-derived myocardium and laryngeal muscles, consistent with a CPM origin (*Lescroart et al., 2015*). Moreover, cucullaris development follows a branchiomeric program and cucullaris-derived muscles were shown to be absent in *Tbx1*-null mice (*Kelly et al., 2004*; *Lescroart et al., 2015*; *Sefton et al., 2016*; *Theis et al., 2010*). Nevertheless, the source of the cucullaris is still equivocal due to the location of its embryonic anlagen at the interface of cranial, somitic and lateral plate mesodermal populations.

Skeletal elements and muscle-associated connective tissue (MCT) also have distinct embryological origins along the rostro-caudal axis. The connective tissue of branchiomeric and tongue muscles originate from neural crest cells (NCCs) of cranial origin (*Evans and Noden, 2006*; *Köntges and Lumsden, 1996*; *Noden, 1983*; *Noden, 1988*; *Ziermann et al., 2018b*). Cranial NCCs also give rise to skeletal components and tendons in the head. In contrast, the skeleton and connective tissue originate from somitic mesoderm in the trunk and from LPM in limbs (*Nassari et al., 2017*). The neck and shoulder girdle contain skeletal elements and connective tissues of distinct NCC, LPM or somitic origins (*Durland et al., 2008*; *Matsuoka et al., 2005*; *McGonnell et al., 2001*; *Nagashima et al., 2016*; *Tabler et al., 2017*; *Valasek et al., 2010*). It has been suggested that NCCs form both connective tissue and endochondral cells at the attachment sites of neck muscles to shoulders in mouse (*Matsuoka et al., 2005*). However, studies in non-mammalian animals have contested a NCC contribution to the pectoral girdle (*Epperlein et al., 2012*; *Kague et al., 2012*; *Ponomartsev et al., 2017*).

Therefore, the neck region consists of muscle, skeletal and connective tissue elements of mixed cellular origins, underscoring the difficulty in assigning embryonic identities for these structures. In addition, the genetic requirements for the formation of non-somitic and somitic neck muscles remain

to be defined. To resolve these issues, we used genetic lineage and loss-of-function mice combined with histology, µCT and 3D reconstructions to map the embryological origins of all neck muscles and associated connective tissues. In doing so, we show that cucullaris-derived muscles originate from a posterior CPM population and are differentially affected in *Tbx1*-null mice. Moreover, we identify a unique genetic network involving both *Mesp1* and *Pax3* genes for somite-derived neck muscles and we define a new limit of neural crest contribution to neck connective tissue and shoulder components.

## Results

### Distinct myogenic programs define neck muscle morphogenesis

To investigate the embryological origin of neck muscles in the mouse, we mapped CPM- and somite-derived myogenic cells using lineage-specific *Cre* drivers including *Mef2c-AHF*, *Islet1*, *Mesp1* and *Pax3* (*Figure 1*). The *Mef2c-AHF* (anterior heart field) enhancer is activated in the second heart field and myogenic progenitors of CPM origin (*Lescroart et al., 2010*; *Verzi et al., 2005*). *Islet1* and *Mesp1* genes are both expressed in early CPM and are essential for cardiac development. The *Mesp1* lineage also marks some anterior somitic derivatives (*Cai et al., 2003*; *Harel et al., 2009*; *Saga et al., 2000*; *Saga et al., 1999*). In contrast, *Pax3* is activated in all somitic progenitors and is a key actor during trunk and limb muscle formation (*Relaix et al., 2005*; *Tajbakhsh et al., 1997*; *Tremblay et al., 1998*). Given that the majority of *Mef2c-AHF* derivatives are myogenic cells (*Lescroart et al., 2015*; *Lescroart et al., 2010*; *Verzi et al., 2005*), we analyzed this lineage using *Rosa26*$^{R-lacZ/+}$ (*R26R*) reporter mice. *Islet1*, *Mesp1* and *Pax3* genes are also expressed in cells contributing to skeletal components, connective tissues or neurons. To focus on the myogenic lineage, we used *Pax7*$^{nGFP-stop/nlacZ}$ (*Pax7*$^{GPL}$) reporter mice, which mark cells with nuclear β-galactosidase (β-gal) activity following *Cre* recombination (*Sambasivan et al., 2013*).

We first examined embryos after myogenic specification (E10.5 and E11.75), and fetuses when muscles are patterned (E18.5). In *Mef2c-AHF*$^{Cre}$;*R26R* embryos, β-gal-positive cells were observed in the mesodermal core of pharyngeal arches at the origin of branchiomeric muscles, in second heart field derivatives, and in the cucullaris anlage (*Figure 1A,E*). A spatiotemporal analysis of the cucullaris using *Myf5*$^{Cre}$;*Pax7*$^{GPL}$ and *Myf5*$^{Cre}$;*R26*$^{mTmG}$ embryos (*Figure 1—figure supplement 1*) showed that *Myf5*-derived muscle progenitors located at the level of the posterior pharyngeal arches, and adjacent to somites S1-S3 (*Figure 1—figure supplement 1A'*), were innervated by the accessory nerve XI (*Figure 1—figure supplement 1G–G''*). These cells gave rise to the trapezius and sternocleidomastoid muscles (*Figure 1—figure supplement 1A–F'*) thus confirming the identity of the cucullaris anlage in mouse (*Tada and Kuratani, 2015*).

In *Islet1*$^{Cre}$;*Pax7*$^{GPL}$ and *Mesp1*$^{Cre}$;*Pax7*$^{GPL}$ embryos, labeling was also observed in pharyngeal arch derivatives and the cucullaris (*Figure 1B–C,F–G*), the latter showing less contribution from the *Islet1* lineage. On sections, a subset of the Myod-positive cells in the cucullaris originated from *Islet1*-derived cells (*Figure 1—figure supplement 2A*). Surprisingly, *Pax3*$^{Cre}$;*Pax7*$^{GPL}$ embryos also showed *lacZ* expression in the cucullaris at E11.75, although no expression was detected at E10.5 (*Figure 1D,H*). Given that *Pax3* and *Pax7* are also expressed in neural crest cells (*Relaix et al., 2004*), and that these *Pax3/Pax7*-derived cells were excluded from the Myod-positive myogenic population at E12.5 after muscle specification (*Figure 1—figure supplement 2B*), they are likely to be of NCC origin. As expected, *Pax3* lineage tracing also labeled the somite-derived myotomes, hypaxial migrating progenitors that form the hypoglossal cord (origin of tongue and infrahyoid muscles), and limb muscle progenitors. Furthermore, the hypaxial anlage, which is located at the proximal limb bud and gives rise to the cutaneous maximus and latissimus dorsi muscles, was *Pax3*-derived (*Figure 1D,H*; *Figure 1—figure supplement 1D'*) (*Prunotto et al., 2004*; *Tremblay et al., 1998*). Unexpectedly, this anlage and the latissimus dorsi muscle were also labeled in *Islet1*$^{Cre}$;*Pax7*$^{GPL}$ but not in *Mesp1*$^{Cre}$;*Pax7*$^{GPL}$ embryos (*Figure 1F–G,J–K*). On sections at E12.5, Islet1 expression was observed in *Pax3*-derived cells after the emergence of myogenic cells from the proximal limb bud (*Figure 1—figure supplement 2C*). In addition, the *Mesp1* lineage contributed to anterior somitic derivatives during early embryonic development as previously reported (*Loebel et al., 2012*; *Saga et al., 1999*); strong *lacZ* expression was observed in the hypoglossal

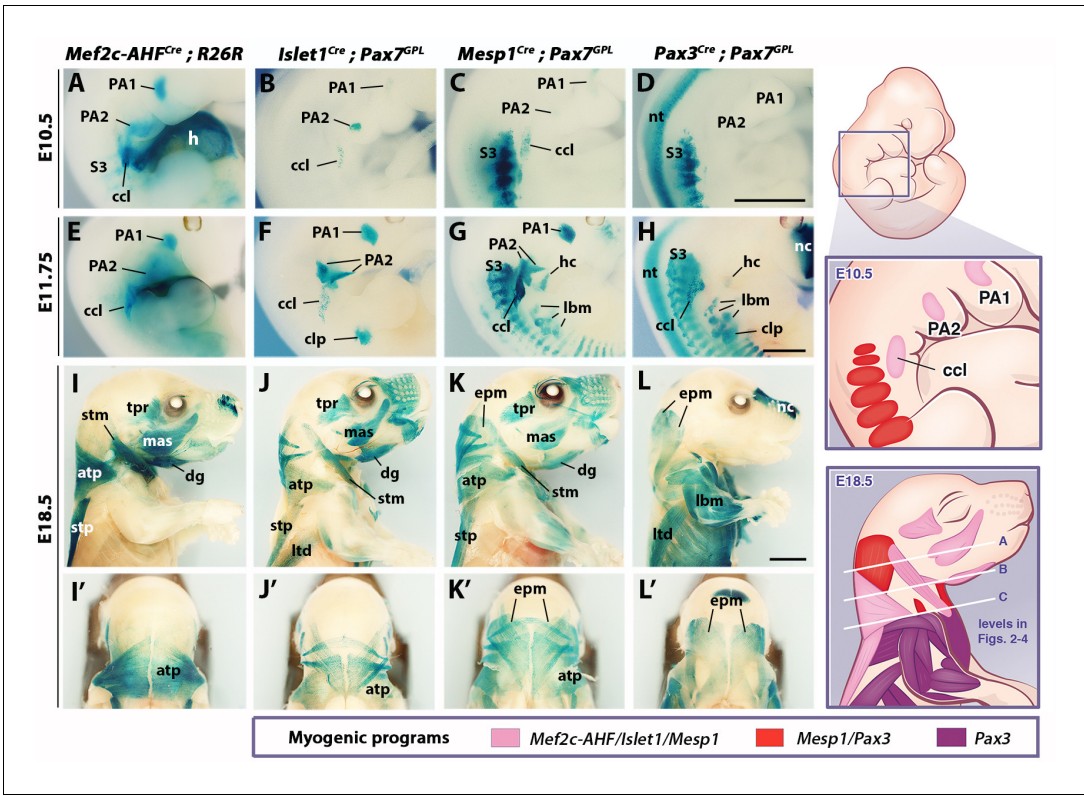

**Figure 1.** Genetic lineage tracing of neck muscle progenitors. Whole-mount X-gal stainings of *Mef2c-AHF^Cre*; *R26R, Islet1^Cre;Pax7^GPL*, *Mesp1^Cre;Pax7^GPL* and *Pax3^Cre;Pax7^GPL* mice at E10.5 (**A–D**), E11.75 (**E–H**) and E18.5 (**I–L'**) (n = 3 for each condition). See associated *Figure 1—supplements 1–3*. (**A–H**) Note labeling of mesodermal core of pharyngeal arches (PAs) and cucullaris anlage (ccl) by *Mef2c-AHF, Islet1* and *Mesp1* lineage reporters; β-gal+ cells in anterior somites of *Mesp1^Cre* embryos and in the *clp* anlagen of *Islet1^Cre* embryos. *Pax3* lineage marked somitic mesoderm. (**I–L'**) *Mef2c-AHF, Islet1* and *Mesp1* lineages marked branchiomeric (mas, tpr, dg) and cucullaris muscles (stm, atp and stp). *Pax3^Cre* and *Mesp1^Cre* labeled somitic epaxial neck muscles (epm). atp, acromiotrapezius; ccl, cucullaris anlage; clp, cutaneous maximus/latissimus dorsi precursor; dg, digastric; epm, epaxial musculature; h, heart; hc, hypoglossal cord; lbm, limb muscle anlagen and limb muscles; ltd, latissimus dorsi; mas, masseter; nc, nasal capsule; nt, neural tube; PA1-2, pharyngeal arches 1–2; S3, somite 3; stm, sternocleidomastoid; stp, spinotrapezius; tpr; temporal. Scale bars: in D for A-D and in H for E-H, 1000 μm; in L for I-L', 2000 μm.

DOI: https://doi.org/10.7554/eLife.40179.002

The following figure supplements are available for figure 1:

**Figure supplement 1.** Ontogenetic analysis of *Myf5* muscle progenitors at the head-trunk interface.
DOI: https://doi.org/10.7554/eLife.40179.003
**Figure supplement 2.** *Mef2c-AHF, Islet1, Mesp1* and *Pax3* lineage tracings using *lacZ* reporters.
DOI: https://doi.org/10.7554/eLife.40179.004
**Figure supplement 3.** *Mesp1* and *Pax3* lineage tracings in somitic neck muscles using the *Pax7^GPL* reporter.
DOI: https://doi.org/10.7554/eLife.40179.005

cord and somites S1-S6. Labeling decreased in more posterior myotomes and in forelimb muscle progenitors compared to *Pax3^Cre;Pax7^GPL* embryos (*Figure 1C–D,G–H*).

Lineage tracings with *Mef2c-AHF^Cre*, *Islet1^Cre* and *Mesp1^Cre* marked branchiomeric (temporal, masseter, digastric, mylohyoid and pharyngeal) and cucullaris-derived neck muscles (acromiotrapezius, spinotrapezius and sternocleidomastoid), all of which were excluded from the *Pax3* lineage (*Figure 1I–L*, *Figure 1—figure supplement 2D–G'*). These findings support previous studies showing that cucullaris muscle development is controlled by a branchiomeric myogenic program (*Kelly et al., 2004*; *Lescroart et al., 2015*; *Sefton et al., 2016*; *Theis et al., 2010*). In addition, both

*Mesp1* and *Pax3* lineages labeled somitic neck muscles (*Figure 1K–L'*, *Figure 1—figure supplement 2F–G'*).

Analysis of different somite-derived neck muscles on sections showed that *Mesp1* and *Pax3* lineages gave rise to the great majority of the Pax7-positive myogenic population (*Figure 1—figure supplement 3*), demonstrating the high recombination efficiency obtained with the *Cre* lines. The results indicate that neck somitic muscles originate from myogenic cells that have expressed both *Mesp1* and *Pax3* genes.

To further investigate the contributions of *Mesp1* and *Pax3* lineages to neck muscles, we examined sections using the $R26^{tdTomato}$ reporter co-immunostained with the myofibre marker Tnnt3 at three representative levels (A, B and C levels in *Figure 1*; see also *Figure 2—figure supplement 1*). At anterior levels, while *Pax3* lineage contribution was limited to somite-derived neck muscles, the *Mesp1* lineage marked branchiomeric muscles (mylohyoid, pharyngeal, laryngeal, esophagus), cucullaris-derived muscles (acromiotrapezius and sternocleidomastoid) and somite-derived neck muscles (*Figure 2A–H*, *Figure 1—figure supplement 2F–G'*, *Figure 2—figure supplement 2A–H'*). The epaxial and hypaxial neck muscles showed equivalent Tomato expression in both $Mesp1^{Cre};R26^{tdTomato}$ and $Pax3^{Cre};R26^{tdTomato}$ mice. These observations further indicate that *Mesp1* and *Pax3* lineages contribute equivalently to neck muscles derived from anterior somites.

At the shoulder level, we observed less *Mesp1* contribution to more posterior somitic muscles (*Figure 2I–J*). In contrast to that observed at anterior levels, little or no Tomato expression was detected in myofibres of scapular muscles in $Mesp1^{Cre};R26^{tdTomato}$ mice (*Figure 2—figure supplement 2I–J'*). Therefore, *Mesp1* lineage contribution was restricted to epaxial and hypaxial neck muscles, in contrast to pectoral and trunk muscles that originate from the *Pax3* lineage (*Figures 1–2*) (*Table 1*). These observations lead us to propose that three distinct myogenic programs are involved in the formation of neck and pectoral musculature at the head-trunk interface. The branchiomeric and cucullaris-derived muscles depend on a common myogenic program involving *Mef2c-AHF*, *Islet1* and *Mesp1* lineages; the somitic neck muscles that originate from anterior somites derive from both *Mesp1* and *Pax3* lineages; the pectoral muscles derived from more posterior somites depend on the activation of *Pax3* only (*Table 1*).

## Dual neural crest and mesodermal origins of neck connective tissues

To define the cellular origin of neck muscle-associated connective tissue (MCT), we traced the contribution of different embryonic populations using $Mesp1^{Cre};R26^{tdTomato}$ and $Pax3^{Cre};R26^{tdTomato}$ mice as well as $Wnt1^{Cre}$ and $Prx1^{Cre}$ reporters that label NCC and postcranial LPM derivatives, respectively (*Burke and Nowicki, 2003*; *Danielian et al., 1998*; *Durland et al., 2008*). Both NCC and LPM populations were reported to contribute to trapezius MCT (*Durland et al., 2008*; *Matsuoka et al., 2005*). Moreover, it was suggested that the postcranial LPM is a source for cucullaris-derived muscles (*Theis et al., 2010*). A direct comparison of NCC and LPM derivatives allowed us to clarify the contribution of these two populations to cucullaris formation (*Figures 3–4*).

We first investigated the distribution of neck muscles and NCCs using $Myf5^{nlacZ/+}$, $Mef2c-AHF^{Cre};R26R$, $Pax3^{Cre};R26R$ and $Wnt1^{Cre};R26R$ embryos (*Figure 3—figure supplement 1*). At E10.5, the cucullaris anlage was positioned at the level of posterior pharyngeal arches where *Wnt1*-derived-positive cells were detectable (*Figure 1A–C*, *Figure 1—figure supplement 1A'*, *Figure 3—figure supplement 1A–B*). Subsequently, the cucullaris progenitors expanded caudally from E11.5 to E13.5. The posterior limit of the cranial NCC domain also extended posteriorly; however, the *Wnt1*-labeled cells did not cover the posterior portion of cucullaris-derived muscles (*Figure 3—figure supplement 1C–H*). At E14.5, the acromiotrapezius and spinotrapezius attained their definitive position in $Myf5^{nlacZ/+}$ and $Mef2c-AHF^{Cre};R26R$ embryos (*Figure 3—figure supplement 1I–J*). *Wnt1*-derived cells were observed in the anterior acromiotrapezius muscle, but not in the spinotrapezius that was situated in a *Pax3*-derived domain (*Figure 3—figure supplement 1K–L*). Analysis of whole-mount embryos indicated that the cranial NCCs did not contribute to connective tissue of posterior cucullaris derivatives, in contrast to what was reported previously (*Matsuoka et al., 2005*).

To further analyze NCC contribution to the cervical region at the cellular level, we performed immunostainings on sections for Tomato and Tnnt3 in E18.5 $Wnt1^{Cre};R26^{tdTomato}$ fetuses (*Figure 3*, *Figure 3—figure supplement 2*). Given that the *Wnt1* lineage is a source of both neuronal and connective tissue derivatives, we associated Tomato immunostaining with Tuj1 that marks neuronal cells and with Tcf4 that labels MCT fibroblasts (*Figure 3*, *Figure 3—figure supplements 2–3*). At the

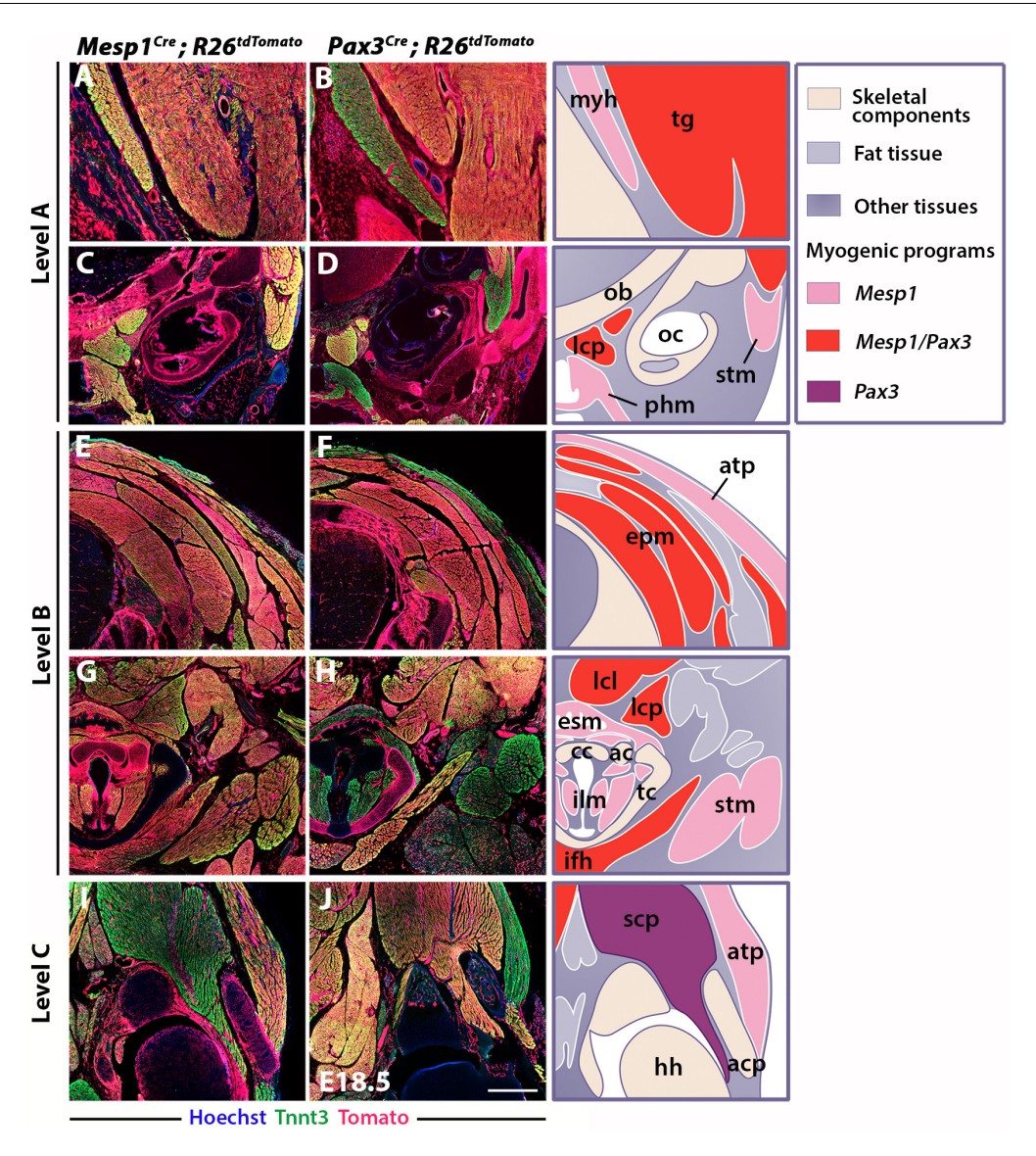

**Figure 2.** Differential contributions of *Mesp1* and *Pax3* lineages to neck and shoulders. Immunostainings on coronal cryosections of E18.5 *Mesp1^Cre^;R26^tdTomato^* and *Pax3^Cre^;R26^tdTomato^* mice for the myofibre Tnnt3 and Tomato markers at levels indicated in *Figure 1*. Higher magnifications of selected areas in (**A–J**) are shown in *Figure 2—figure supplement 2*; (n = 2 for each condition). See also the atlas of neck musculature in *Figure 2—figure supplement 1*. (**A–H**) *Mesp1^Cre^* labeled all neck muscles including branchiomeric (myh, esm, phm and ilm), cucullaris (stm, atp), somitic epaxial (epm) and hypaxial (tg, lcp, lcl, ifh) muscles. *Pax3^Cre^* marked somitic muscles. (**I–J**) At shoulder level, *Mesp1*-derived cells did not contribute to posterior somitic myofibres including scapular muscles (scp) compared to that observed in *Pax3^Cre^* embryos. ac, arytenoid cartilage; acp, scapular acromion process; atp, acromiotrapezius; cc, cricoid cartilage; epm, epaxial musculature; esm, esophagus striated muscle; hh, humeral head; ifh, infrahyoid muscles; ilm, intrinsic laryngeal muscles; lcl, longus colli; lcp, longus capitis; myh, mylohyoid; ob, occipital bone; oc, otic capsule; phm, pharyngeal muscles; stm, sternocleidomastoid; scp, scapular musculature; tc, thyroid cartilage; tg, tongue. Scale bars: in J for A-B 200 μm, for C-J 400 μm.
DOI: https://doi.org/10.7554/eLife.40179.006

The following figure supplements are available for figure 2:

**Figure supplement 1.** Atlas of neck musculature in mouse.
DOI: https://doi.org/10.7554/eLife.40179.007
**Figure supplement 2.** *Mesp1* and *Pax3* lineage contributions to neck and shoulder muscles.
DOI: https://doi.org/10.7554/eLife.40179.008

**Table 1.** Contribution of *Mef2c-AHF*, *Islet1*, *Mesp1* and *Pax3* lineages to neck and pectoral musculature.

| *Mef2c/Islet1/Mesp1*-derived muscles | *Mesp1/Pax3*-derived muscles | *Pax3*- derived muscles |
|---|---|---|
| Mylohyoid<br>Digastric muscles | Epaxial neck muscles (splenius, semispinalis, levator scapula, rhomboid occipitalis, suboccipital and postvertebral muscles) | Scapular muscles (supraspinatus, Infraspinatus, subscapularis) |
| Pharyngeal muscles<br>Intrinsic laryngeal muscles<br>Esophagus striated muscle | | Pectoralis |
| | | Latissimus dorsi[†]<br>Cutaneous maximus[†] |
| Sternocleidomastoid<br>Acromiotrapezius<br>Spinotrapezius | Hypaxial neck muscles (tongue muscles*, infrahyoid muscles, longus capitis, longus colli) | |
| Branchiomeric myogenic program | Anterior-most somite myogenic program | More posterior somite myogenic program |

*Including intrinsic and extrinsic tongue muscles of somitic origin

[†]Also derived from an *Islet1* lineage

DOI: https://doi.org/10.7554/eLife.40179.009

cranial level, the MCT of branchiomeric (masseter, mylohyoid), tongue and acromiotrapezius muscles was derived from *Wnt1*- and *Pax3*-lineages but not from the mesodermal *Mesp1* lineage (*Figure 3—figure supplement 2A–B'*, *Figure 3—figure supplement 3A,F*, *Figure 3—figure supplement 4A–D,G*). The acromiotrapezius showed a high contribution from *Wnt1*-derived cells while the underlying epaxial muscles had considerably less labeled cells that were limited to the neuronal Tuj1-positive population (*Figure 3A–A'*). The *Wnt1* lineage gave rise to Tcf4-positive fibroblasts in the acromiotrapezius, but not in epaxial neck muscles, where fibroblasts were derived from the *Mesp1* lineage (*Figure 3—figure supplements 3B–C* and *4E*). These observations are in accordance with a NCC origin of branchiomeric, anterior trapezius and tongue connective tissue as reported previously (*Matsuoka et al., 2005*).

However, the NCC contribution to connective tissue in the sternocleidomastoid subset of cucullaris-derived muscles appeared more heterogeneous than that observed in the acromiotrapezius. In rodents, the sternocleidomastoid is composed of three individual muscles (cleidomastoid, sternomastoid and cleido-occipitalis portions); a differential NCC contribution to MCT was observed in these muscles. While *Wnt1*-derived NCCs were widely present in the sternomastoid and cleido-occipitalis, the NCC contribution was limited in the cleidomastoid (*Figure 3B–B'*). Indeed, Tcf4-positive fibroblasts in the cleido-occipitalis originated from the *Wnt1* lineage, whereas the majority of MCT fibroblasts in the cleidomastoid were derived from the *Mesp1* lineage (*Figure 3—figure supplements 3D–E* and *4F*).

A differential contribution of NCCs to connective tissue was also seen within the laryngeal and infrahyoid musculature. Extensive *Wnt1* lineage contributions to MCT was observed in laryngeal muscles (thyroarytenoid and cricothyroid) that connect to the thyroid cartilage, which is of NCC origin (*Figure 3C–C'*) (*Tabler et al., 2017*). In contrast, the laryngeal muscles (cricoarytenoid and vocal muscles) that link mesoderm-derived laryngeal cartilages (cricoid, arytenoid and medio-caudal portion of the thyroid) did not contain NCC-derived connective tissue (*Figures 2G–H* and *3C–C'*) (*Tabler et al., 2017*). In these muscles, the *Wnt1*-derived cells were neuronal, as observed in the esophagus, whereas the MCT fibroblasts were derived from the *Mesp1* lineage (*Figure 3C–C'*, *Figure 3—figure supplements 2D–D'* and *4H*). As another example, *Wnt1*-derived cells contributed to a greater extent to MCT in infrahyoid muscles (thyrohyoid muscles) that connect the hyoid and thyroid cartilage that are of NCC origin, compared to infrahyoid muscles (omohyoid and sternohyoid muscles) that link posteriorly pectoral structures of mesodermal origin (*Figure 3—figure supplement 2C,C', H*; *Figure 3—figure supplement 3G–H*). These observations suggest that MCT composition within laryngeal and infrahyoid muscles correlates in part with the embryonic origin of the skeletal components to which they attach (*Figure 2G–H*, *Figure 3C–C'*, *Figure 3—figure supplement 2C–C', H*).

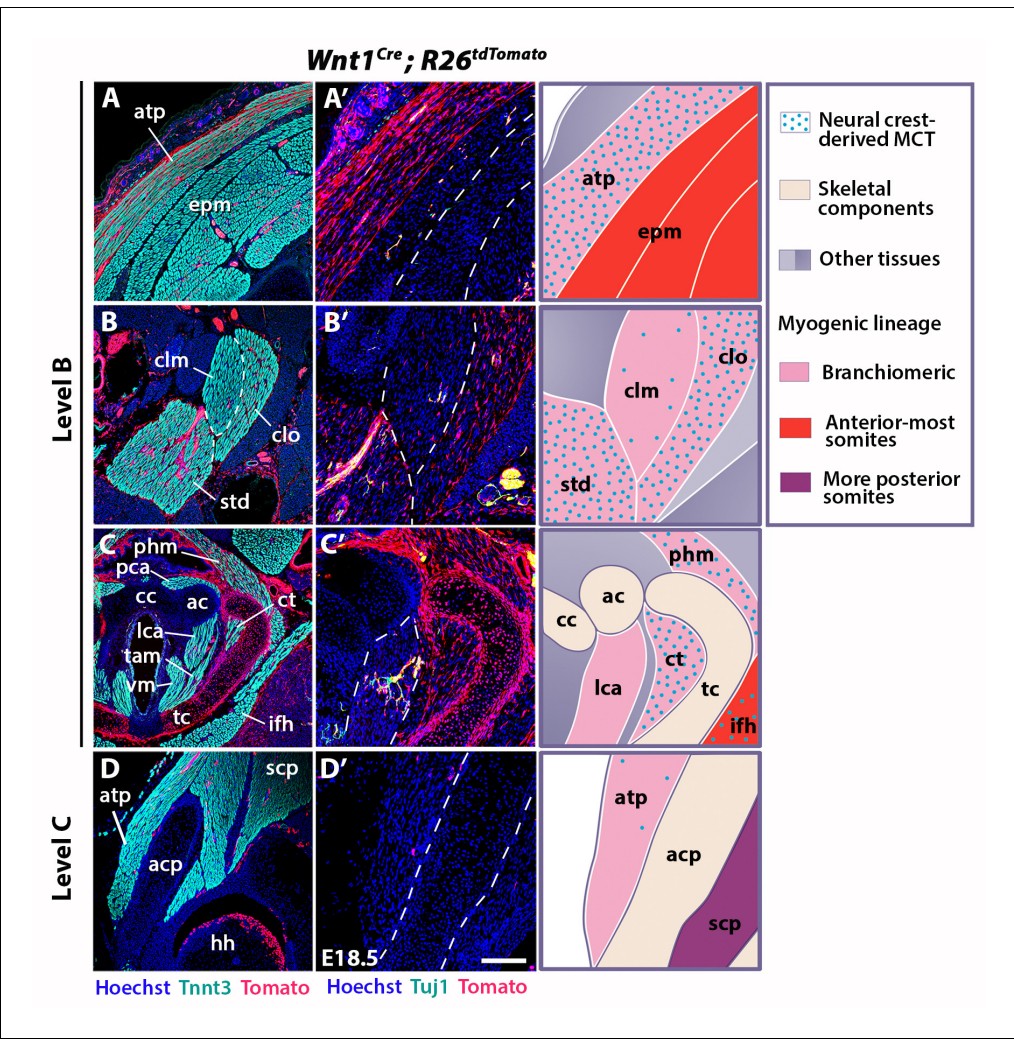

**Figure 3.** Neural crest contribution to neck muscle-associated tissue. Immunostainings on coronal cryosections of E18.5 *Wnt1^Cre;R26^tdTomato* mice at levels indicated in *Figure 1*. Tnnt3/Tomato immunostainings are shown in (**A–D**) and immunostainings for Tuj1/Tomato on selected areas of (**A–D**) are shown with higher magnifications in (**A'–D'**). See associated *Figure 3—figure supplement 1–4*; (n = 2). (**A–A'**) Note high *Wnt1* contribution in the acromiotrapezius but not in epaxial muscles where *Wnt1*-derived cells marked neuronal cells. (**B–C'**) *Wnt1*-derived cells marked differentially the distinct muscles composing the sternocleidomastoid and laryngeal musculatures. (**D–D'**) At shoulder level, the *Wnt1* cells did not contribute to attachment of acromiotrapezius to scapula. ac, arytenoid cartilage; acp, scapular acromion process; atp, acromiotrapezius; cc, cricoid cartilage; clm, cleidomastoid; clo, cleido-occipitalis; ct, cricothyroid; epm, epaxial musculature; hh, humeral head; ifh, infrahyoid muscles; lca, lateral cricoarytenoid; MCT, muscle-associated connective tissue; pca, posterior cricoarytenoid; phm, pharyngeal muscles; scp, scapular musculature; std, sternomastoid; tam, thyroarytenoid muscle; tc, thyroid cartilage; vm, vocal muscle. Scale bars: in D' for A-D 400 µm for A'-D' 200 µm.

DOI: https://doi.org/10.7554/eLife.40179.010

The following figure supplements are available for figure 3:

**Figure supplement 1.** Distribution of developing neck muscles and neural crest cells.
DOI: https://doi.org/10.7554/eLife.40179.011

**Figure supplement 2.** Neural crest contribution to neck and pectoral structures.
DOI: https://doi.org/10.7554/eLife.40179.012

**Figure supplement 3.** *Wnt1* lineage contribution to connective tissue fibroblasts.
DOI: https://doi.org/10.7554/eLife.40179.013

**Figure supplement 4.** Contribution of *Pax3* and *Mesp1* lineages to connective tissue fibroblasts.
DOI: https://doi.org/10.7554/eLife.40179.014

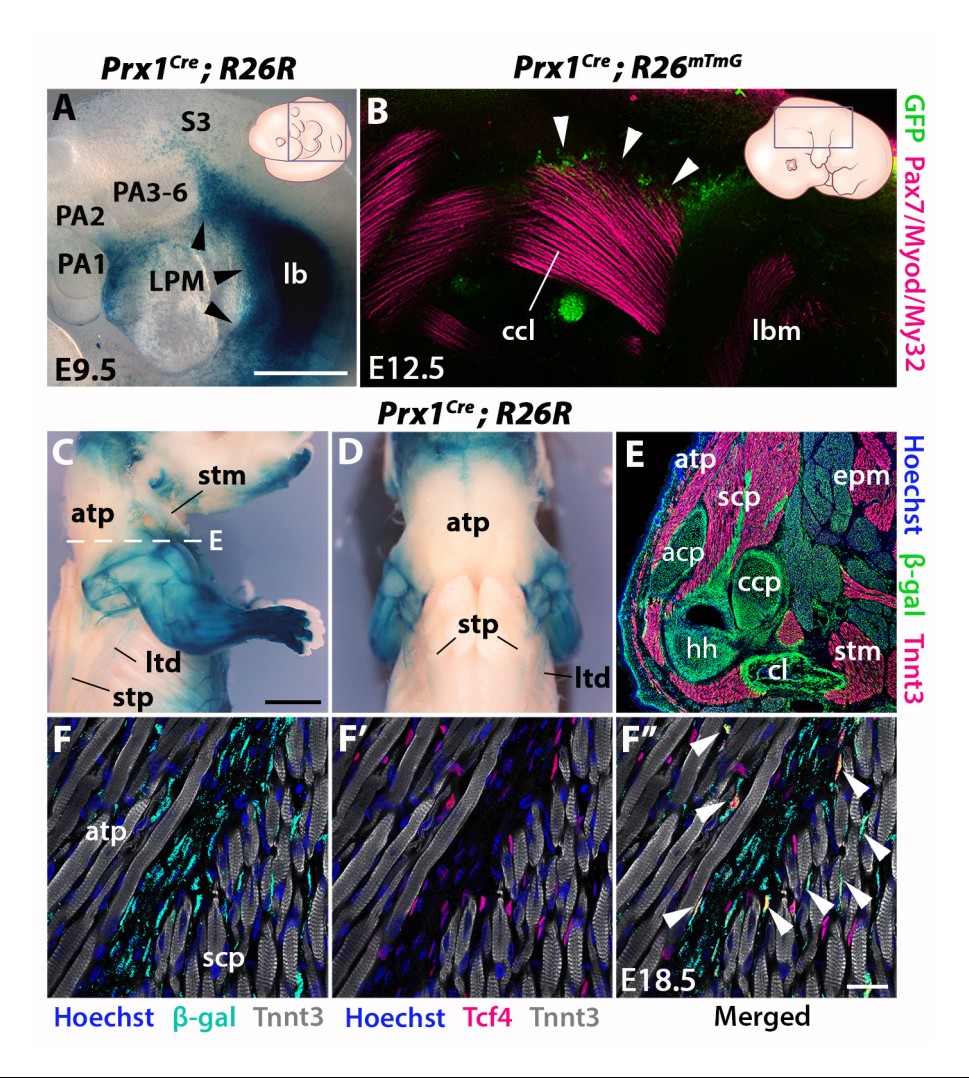

**Figure 4.** *Prx1*-LPM lineage contribution to neck and pectoral girdle. See also *Figure 4—figure supplement 1, 2*. (A–D) X-gal stainings of *Prx1^Cre^;R26R* reporter mice at E9.5 (n = 3) (A) and E18.5 (n = 3) (C–D), and immunostaining for GFP and the Pax7/Myod/My32 myogenic markers in *Prx1^Cre^;R26^mTmG* E12.5 embryo (n = 2) (B). Note *Prx1*-derived cells in postcranial LPM (A, black arrowheads) and *Prx1*-derived cells among, but not in, cucullaris myofibres (B–D). (E–F'') Immunostaining for β-gal, Tnnt3 and Tcf4 on coronal cryosections of E18.5 *Prx1^Cre^;R26R* mice (n = 2) showed β-gal⁺ cells constituting the pectoral girdle (E, level C in *Figure 1*) and in MCT fibroblasts (F-F'', white arrowheads), but not in trapezius myofibres. acp, scapular acromion process; atp, acromiotrapezius; ccl, cucullaris anlage; ccp, scapular coracoid process; cl, clavicle; epm, epaxial musculature; hh, humeral head; lb, forelimb bud; lbm, limb muscle anlagen; LPM, lateral plate mesoderm; PA1-6, pharyngeal arches 1–6; S3, somite 3; scp, scapular muscles; stm, sternocleidomastoid; stp, spinotrapezius. Scale bars: in A for A, B 500 μm; in C for C-D 2000 μm, for E 500 μm; in F'' for F-F'' 20 μm.
DOI: https://doi.org/10.7554/eLife.40179.015

The following figure supplements are available for figure 4:

**Figure supplement 1.** Comparison of the *Myf5* and *Prx1* lineage tracings.
DOI: https://doi.org/10.7554/eLife.40179.016

**Figure supplement 2.** *Prx1* lineage contribution to neck and limbs.
DOI: https://doi.org/10.7554/eLife.40179.017

Given our findings that connective tissues of neck muscles have differential contributions of NCC and mesodermal populations, we analyzed the caudal connections of the cucullaris-derived muscles to the pectoral girdle (*Figure 3D*, *Figure 3—figure supplement 2E–H*). The acromiotrapezius

attaches dorsally to the nuchal ligament and ventrally to the scapular acromion process in continuity with the scapular spine. While *Wnt1*-derived cells were present dorsally (*Figure 3A*, *Figure 3—figure supplement 2E*), this contribution diminished gradually and was undetectable at the insertion on the scapula (*Figure 3D–D'*, *Figure 3—figure supplement 2F*). Similarly, the sternocleidomastoid muscle showed limited NCC contribution to the attachment sites of the clavicle and sternum (*Figure 3—figure supplement 2G–H*). In contrast to what was previously described (*Matsuoka et al., 2005*), we did not observe NCC contribution to the shoulder endochondral tissue nor to the nuchal ligament (*Figure 3—figure supplement 2E–H*). Taken together, these observations define a novel boundary for neural crest contribution to neck/pectoral components. The posterior contribution limit of neural crest to branchiomeric MCT occurs at the level of laryngeal muscles that connect to NCC skeletal derivatives. Moreover, NCCs do not participate in connecting posterior cucullaris and infrahyoid muscles to their skeletal elements.

To assess the cellular origin of cucullaris connective tissue at posterior attachment sites, we next traced the contribution of lateral plate mesoderm (LPM) to the neck/shoulder region using *Prx1^{Cre}* reporter mice (*Durland et al., 2008*; *Logan et al., 2002*) (*Figure 4*, *Figure 4—figure supplements 1–2*). Analysis of E9.5 embryos showed that *Prx1*-derived cells contribute to the forelimb bud and cells adjacent to the anterior-most somites, but not to pharyngeal arches (*Figure 4A*). At E12.5, the postcranial *Prx1*-derived domain clearly defined the lateral somitic frontier along the rostrocaudal axis (*Durland et al., 2008*) and did not include the cucullaris anlage (*Figure 4—figure supplement 1*, white arrowheads). Whole-mount immunostainings for the myogenic markers Pax7/Myod/My32 and for GFP in *Prx1^{Cre}*;*R26^{mTmG}* embryos showed that *Prx1*-derived cells were present in the dorsal part of the cucullaris but did not contribute to myofibres (*Figure 4B*, white arrowheads). At E18.5, the *Prx1* lineage marked the limb, scapular and abdominal regions, whereas only a few *Prx1*-derived cells were detected in the cucullaris-derived sternocleidomastoid, acromiotrapezius and spinotrapezius muscles (*Figure 4C–D*). On sections, immunostaining for β-gal and Tnnt3 showed that *Prx1*-derived LPM contributed to limb/shoulder MCT and to skeletal components of the pectoral girdle (*Figure 4E*, *Figure 4—figure supplement 2A–B*). In contrast, fewer *Prx1*-derived cells were detected in the acromiotrapezius and little or no contribution was observed in the epaxial muscles (*Figure 4E*, *Figure 4—figure supplement 2B–C*). In addition, only a limited number of *Prx1*-derived cells gave rise to Tcf4-positive fibroblasts in the trapezius muscles, but they contributed more extensively to the fibroblast population in scapular muscles (*Figure 4F–F''*, white arrowheads, *Figure 4—figure supplement 2D–D"*). Notably, β-gal expression for this lineage was not detected in trapezius myofibres thereby confirming the results obtained at E12.5 (*Figure 4B–F"*, *Figure 4—figure supplements 1–2*).

Therefore, these observations reveal a dual NCC/LPM origin of trapezius connective tissue, with a decrease of NCC contribution at posterior attachment sites. Moreover, our analysis shows that the postcranial LPM does not give rise to cucullaris myofibres in contrast to what was suggested previously (*Theis et al., 2010*), thus providing further evidence for a branchiomeric origin of the cucullaris.

## Divergent functions of *Tbx1* and *Pax3* in neck development

Given the key role for *Tbx1* and *Pax3* genes in the specification of the CPM and somites respectively, we analyzed the effect of inactivation of these genes on neck muscle formation, compared to the muscle phenotypes observed at cranial and trunk levels.

Analysis has been performed by immunostainings on sections and 3D reconstructions of the neck and pectoral girdle using high-resolution micro-computed tomographic (μCT) scans of control, *Tbx1^{-/-}* and *Pax3^{-/-}* fetuses (*Figures 5–6*).

In the early embryo, *Tbx1* is expressed in pharyngeal mesoderm and is required for proper branchiomeric muscle formation (*Grifone et al., 2008*; *Kelly et al., 2004*). While *Tbx1* is expressed in other cranial populations including the pharyngeal ectoderm and endoderm (*Arnold et al., 2006*; *Huynh et al., 2007*), the gene is known to be required cell autonomously during CPM myogenesis (*Kong et al., 2014*; *Zhang et al., 2006*). Analysis of *Tbx1* mutants revealed unexpected features in cucullaris and hypaxial neck muscle formation. As previously described (*Gopalakrishnan et al., 2015*; *Kelly et al., 2004*), anterior branchiomeric muscles (digastric and mylohyoid) showed phenotypic variations, whereas posterior branchiomeric muscles (esophagus and intrinsic laryngeal muscles) and the acromiotrapezius were severely affected or undetectable (*Figure 5B,E,H*;

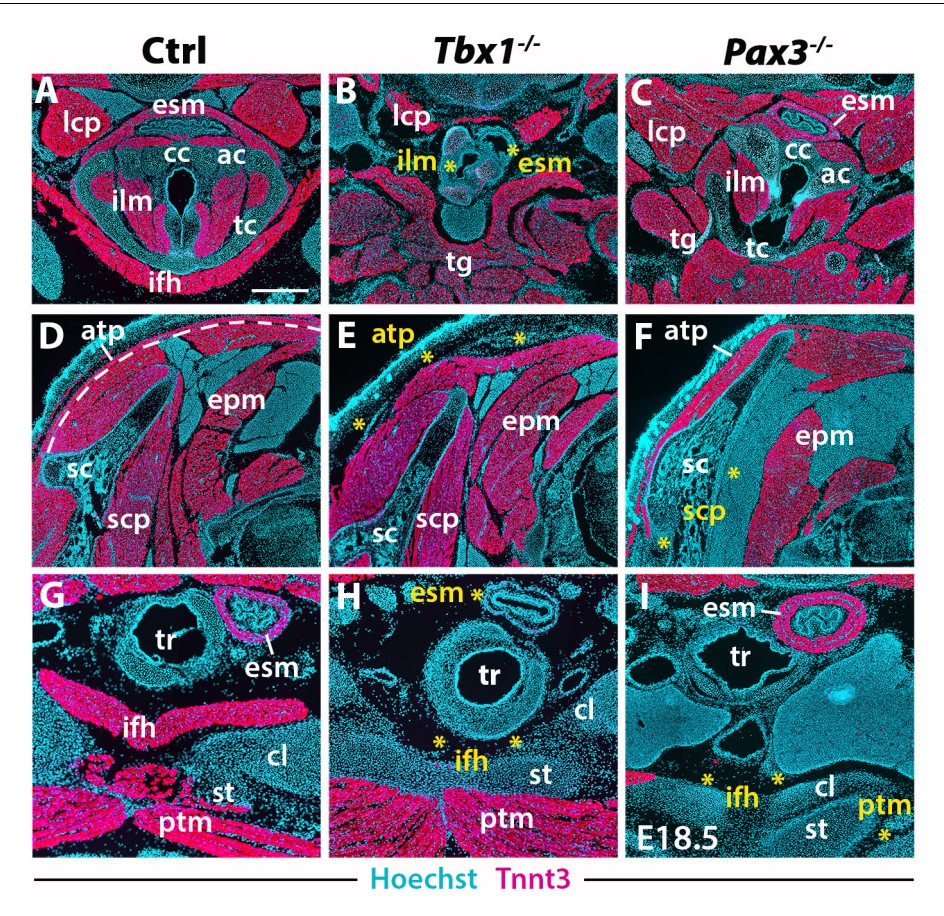

**Figure 5.** Neck muscle phenotypes in *Tbx1* and *Pax3* mutants. (**A–I**) Immunostainings for Tnnt3 on coronal cryosections of control, *Tbx1-null* and *Pax3-null* fetuses at E18.5 (n = 3 each condition). Yellow asterisks indicate missing muscles. Note absence of branchiomeric laryngeal (ilm), esophagus (esm) and trapezius (atp) muscles and severe alteration of somitic infrahyoid muscles (ifh) in *Tbx1* mutants. Scapular (scp) and pectoral (ptm) muscles are missing in *Pax3* mutants. ac, arytenoid cartilage; atp, acromiotrapezius; cc, cricoid cartilage; cl, clavicle; epm, epaxial musculature; esm, esophagus striated muscle; ifh, infrahyoid muscles; ilm, intrinsic laryngeal muscles; lcp, longus capitis; ptm, pectoralis muscles; sc, scapula; scp, scapular muscles; st, sternum; tc, thyroid cartilage; tg, tongue. Scale bars: in A for A-I 500 μm.

DOI: https://doi.org/10.7554/eLife.40179.018

*Figure 6B*) (*Table 2*). However, detailed examination of the cucullaris-derived muscles revealed a heterogeneous dependence on *Tbx1* function that was not reported previously (*Lescroart et al., 2015*; *Theis et al., 2010*). Unexpectedly, the sternocleidomastoid muscle was present bilaterally but smaller (*Figure 6B*); the different portions (cleido-occipitalis, cleidomastoid and sternomastoid) were unilaterally or bilaterally affected in a stochastic manner. Moreover, while the epaxial neck and scapular muscles were unaffected (*Figure 5E*, *Figure 6E–H*), the hypaxial neck muscles derived from anterior somites were altered. Indeed, the tongue and longus capitis were reduced and the infrahyoid and longus colli muscles were severely affected or undetectable (*Figure 5B,H*, *Figure 6E,H*; see interactive 3D PDFs in *Supplementary file 1–2*).

Analysis of *Pax3* mutants showed that the neck and pectoral muscles were differentially affected. As expected, branchiomeric and epaxial muscles developed normally but displayed morphological differences adapted to malformations noted in some skeletal components (*Figure 5C,F*; *Figure 6C, I*). However, whereas hypaxial trunk/limb muscles were severely affected or undetectable in *Pax3* mutants (*Figure 5F,I*; *Figure 6F,I*) (*Tajbakhsh et al., 1997*; *Tremblay et al., 1998*), surprisingly the majority of hypaxial neck muscles derived from both *Mesp1* and *Pax3* lineages were present. Tongue muscles were reduced in size but patterned, the infrahyoid were hypoplastic, whereas the longus

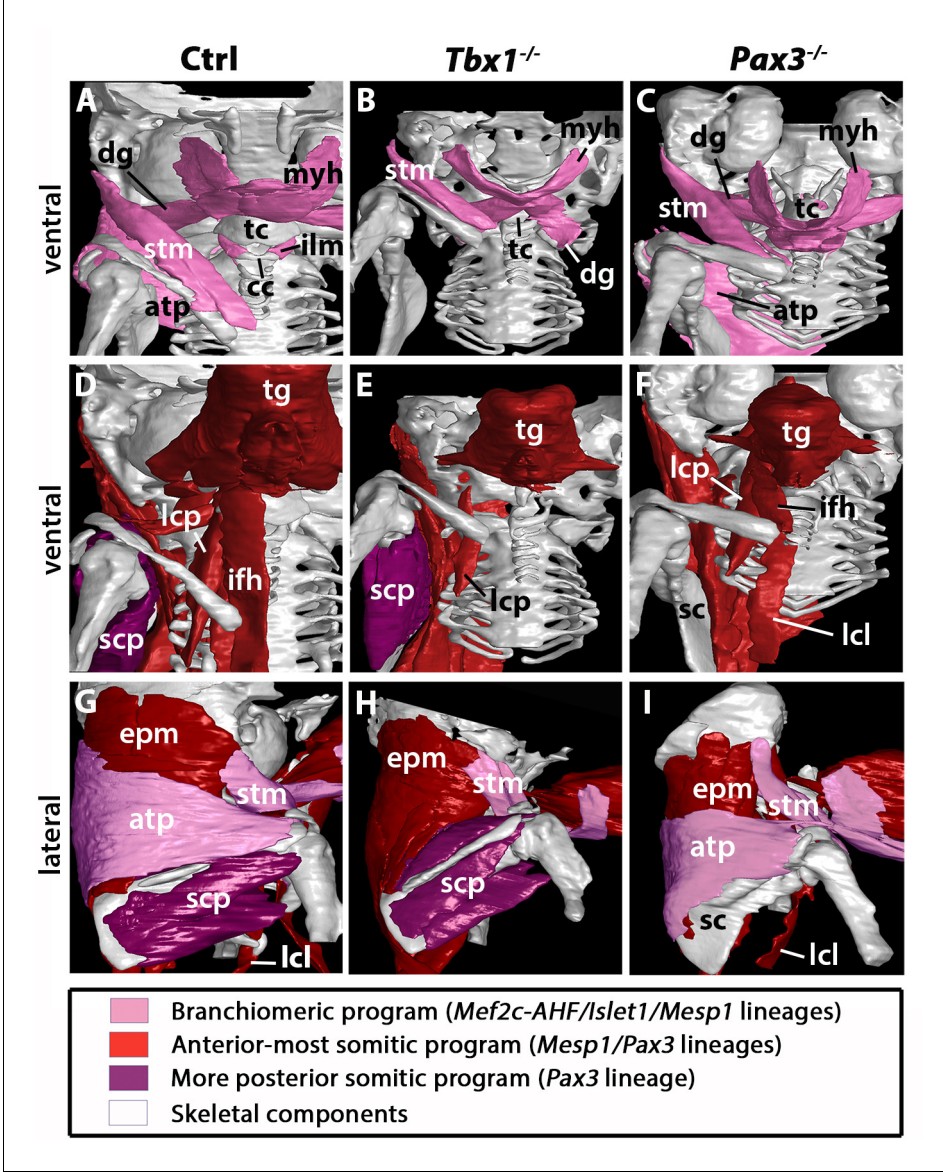

**Figure 6.** 3D reconstructions of neck musculoskeletal system in *Tbx1* and *Pax3* mutants. See interactive 3D PDFs in *Supplementary file 1–3*; control n = 1; mutants n = 2. (**A–C**) Branchiomeric and cucullaris-derived muscles marked by *Mef2c-AHF/Islet1/Mesp1* lineages are indicated in pink. (**D–F**) Anterior somitic muscles (*Mesp1, Pax3* lineages), in red. (**G–I**) Scapular muscles from more posterior somites (*Pax3* lineage), in violet. atp, acromiotrapezius; cc, cricoid cartilage; dg, digastric muscles; epm, epaxial musculature; ifh, infrahyoid muscles; ilm, intrinsic laryngeal muscles; lcl, longus colli; lcp, longus capitis; myh, mylohyoid; sc, scapula; scp, scapular muscles; stm, sternocleidomastoid; tc, thyroid cartilage; tg, tongue.

DOI: https://doi.org/10.7554/eLife.40179.019

capitis and longus colli were unaffected (*Figure 5C*; *Figure 6F,I*; see interactive 3D PDF in *Supplementary file 3*). The phenotypes of the different muscle groups observed in *Tbx1* and *Pax3* mutants are summarized in *Table 2* (see also *Figure 7—figure supplement 1*).

Taken together, these observations revealed that hypaxial muscles in the neck were less affected in *Pax3* mutants than more posterior hypaxial muscles, pointing to distinct requirements for *Pax3* function during neck and trunk muscle formation. In addition, *Tbx1* mutants exhibited more severe phenotypes in hypaxial neck muscles, thus highlighting distinct roles for this gene in branchiomeric and hypaxial neck myogenesis.

**Table 2.** Summary of the neck muscle phenotype observed in *Tbx1*- and *Pax3*-null fetuses.

| | *Tbx1*-null | *Pax3*-null |
|---|---|---|
| **Branchiomeric muscles (*Mef2c-AHF/Islet1/Mesp1* lineage)** | | |
| Mylohyoid | +/- | ++ |
| Digastric muscles | +/- | ++ |
| Intrinsic laryngeal muscles | − | + |
| Esophagus striated muscle | − | ++ |
| Sternocleidomastoid | +/- | + |
| Acromiotrapezius | − | + |
| **Anterior-most somite muscles (*Mesp1/Pax3* lineage)** | | |
| Epaxial musculature | ++ | + |
| Longus capitis | +/- | ++ |
| Longus colli | − | ++ |
| Infrahyoid muscles | − | +/- |
| Tongue muscles* | + | + |
| **More posterior somite muscles (*Pax3* lineage)** | | |
| Scapular muscles | ++ | − |
| Pectoralis | ++ | − |

++,normal; +, altered morphology; +/-, affected; -, severely affected or undetectable

*Including intrinsic and extrinsic tongue muscles of somitic origin

DOI: https://doi.org/10.7554/eLife.40179.020

## Discussion

The embryological origins of neck muscles and connective tissues at the head-trunk interface have been poorly defined largely due to their localization at a transition zone that involves multiple embryonic populations. Using a combination of complementary genetically modified mice and 3D analysis that identifies muscles in the context of their bone attachments, we provide a detailed map of neck tissue morphogenesis and reveal some unexpected features regarding the muscle and connective tissue network.

### Branchiomeric origin of cucullaris-derived muscles

The mammalian neck consists of somitic epaxial/hypaxial muscles, branchiomeric muscles and cucullaris-derived muscles (*Table 1*). The latter constitute a major innovation in vertebrate history, connecting the head to the pectoral girdle in gnathostomes and allowing head mobility in tetrapods (*Ericsson et al., 2013*). Recent studies in different organisms including shark, lungfish and amphibians suggest that the cucullaris develops in series with posterior branchial muscles and that its developmental origin and innervation is conserved among gnathostomes (*Diogo, 2010*; *Ericsson et al., 2013*; *Naumann et al., 2017*; *Noda et al., 2017*; *Sefton et al., 2016*; *Tada and Kuratani, 2015*; *Ziermann et al., 2018a*; *Ziermann et al., 2017*). However, multiple embryological origins including CPM, LPM and somites have been reported for the cucullaris, underscoring the difficulty in deciphering the morphogenesis of this and other muscles in the head-trunk transition zone (*Huang et al., 2000*; *Nagashima et al., 2016*; *Sefton et al., 2016*; *Theis et al., 2010*).

Our study shows that the cucullaris anlage is innervated by the accessory nerve XI and develops contiguously with the mesodermal core of posterior arches and anterior-most somites 1–3. Our lineage analysis reveals that cucullaris development depends on a branchiomeric myogenic program involving *Mef2c-AHF*, *Islet1* and *Mesp1* lineages in keeping with previous results (*Table 1*) (*Lescroart et al., 2015*; *Sefton et al., 2016*; *Theis et al., 2010*). However, our detailed functional analysis and 3D reconstructions lead us to modify the view of the genetic requirements of cucullaris-derived muscles (*Lescroart et al., 2015*; *Theis et al., 2010*). Notably, these muscles are differentially affected in *Tbx1*-null fetuses; the acromiotrapezius does not form while the sternocleidomastoid is present but reduced. Therefore, *Tbx1* is differentially required for sternocleidomastoid and trapezius formation, suggesting that distinct subprograms regulate cucullaris development.

We also demonstrate that the cucullaris anlage is excluded from the postcranial *Prx1*-derived expression domain, which delineates the trunk LPM field (*Figure 4*). The *Prx1* lineage instead gives rise to connective tissue, thereby excluding a contribution from LPM to cucullaris-derived myofibres. Thus, our results, combined with innervation studies, retrospective clonal analyses and grafting experiments in chick and axolotl (*Lescroart et al., 2015*; *Nagashima et al., 2016*; *Sefton et al.,*

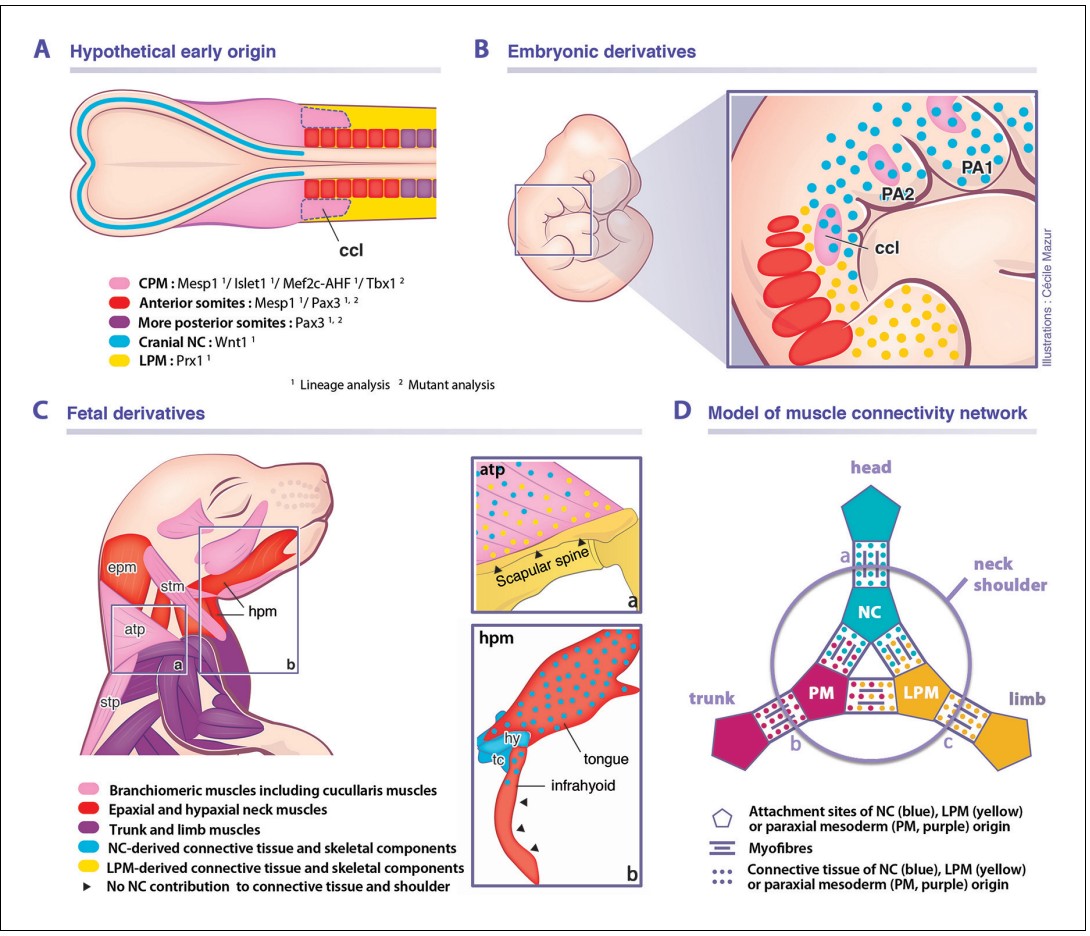

**Figure 7.** Model for musculoskeletal and connective tissue relationships during murine neck development. See also *Figure 7—figure supplement 1*. (A, C) CPM (pink), anterior somites (red) and more posterior somites (violet) muscles are defined by three distinct myogenic programs. (B) Note that the cucullaris develops in a NC domain (blue dots), but is excluded from the postcranial LPM (yellow dots). (C) Dual NC/LPM origin of trapezius connective tissue is indicated in (a). NC contribution to connective tissue extends to tongue and anterior infrahyoid musculature (b). (D) Mixed origins of muscle connective tissues at the head-trunk-limb interface. Example of representative muscles: (a) masseter, (b) spinalis dorsi, (c) deltoid. atp, acromiotrapezius; ccl, cucullaris; CPM, cardiopharyngeal mesoderm; epm, epaxial neck musculature; hpm, hypaxial neck musculature; hy, hyoid bone; LPM, postcranial lateral plate mesoderm; NC, neural crest; PA1-2, pharyngeal arches 1–2; PM, paraxial mesoderm; stm, sternocleidomastoid; stp, spinotrapezius; tc, thyroid cartilage.

DOI: https://doi.org/10.7554/eLife.40179.021

The following figure supplement is available for figure 7:

**Figure supplement 1.** Muscles affected in *Tbx1* and *Pax3* mutants.

DOI: https://doi.org/10.7554/eLife.40179.022

*2016*; *Tada and Kuratani, 2015*), suggest that the postcranial extension of the CPM lateral to the first three somites in tetrapod embryos is a source of cucullaris myogenic cells (*Figure 7A*). The discordance with previous studies regarding the origin of the cucullaris is likely due to its proximity to both anterior somites and LPM (*Figure 7A–B*), and consequently, to potential contamination of embryonic sources in grafting experiments (*Couly et al., 1993*; *Huang et al., 1997*; *Huang et al., 2000*; *Noden, 1983*; *Piekarski and Olsson, 2007*; *Theis et al., 2010*).

## A unique genetic program for somite-derived neck muscles

Our study also points to a unique *Mesp1/Pax3* genetic program in anterior somites for epaxial/hypaxial neck muscle formation (*Table 1*). While it was shown that the *Mesp1* lineage gives rise to tongue muscles (*Harel et al., 2009*), we demonstrate that it also contributes to all neck muscles. In

chordates, *Mesp* genes appear to play a conserved role in cardiogenesis and axis segmentation. In mouse, *Mesp1* inactivation causes early embryonic death from abnormal heart development, and *Mesp1/Mesp2* double-knockout embryos lack non-axial mesoderm (*Moreno et al., 2008*; *Saga, 1998*; *Saga et al., 2000*; *Satou et al., 2004*; *Sawada et al., 2000*). During early murine development, *Mesp1* shows two waves of activation; initially in the nascent mesoderm destined for extra-embryonic, cranial and cardiac mesoderm at the onset of gastrulation; later during somitogenesis, transient *Mesp1* expression is limited to anterior presomic mesoderm (*Saga, 1998*; *Saga et al., 1996*; *Saga et al., 2000*; *Saga et al., 1999*). Our lineage analysis shows that *Mesp1* extensively labels the anterior mesoderm, including the CPM and anterior somites 1–6, while contribution decreases in more posterior somites (*Figure 1*) (*Loebel et al., 2012*; *Saga et al., 2000*; *Saga et al., 1999*). Previous fate mapping experiments have shown that the mesoderm of late-streak stage embryos contributes to both CPM and anterior somites (*Parameswaran and Tam, 1995*). It appears that the first wave of *Mesp1* expression defines not only the CPM field but also includes the mesoderm destined for anterior somites. In contrast, the *Mesp1*-labeled cells observed in more posterior somites using the sensitive *Pax7$^{GPL}$* reporter may result from the transient wave of *Mesp1* expression in the presomic mesoderm during axis segmentation. Furthermore, we show that *Mesp1*-derived anterior somites give rise to all epaxial/hypaxial neck muscles in contrast to trunk/limb muscles originating from more posterior somites marked by *Pax3*. The boundary of *Mesp1* lineage contribution to muscles corresponds to the neck/pectoral interface. Our findings indicate that the anterior somitic mesoderm employs a specific transition program for neck muscle formation involving both *Mesp1* and *Pax3* genes implicated in CPM and somitic myogenesis, respectively (*Figure 7A–C*).

Whereas little is known about the function of *Mesp* genes in chordates, there is evidence that *Mesp1* might be differentially required during anterior *versus* posterior somitic formation. In mouse, different *Mesp1* enhancer activities have been observed between CPM/anterior somites and posterior somites indicating that the regulation of *Mesp1* expression might differ in the two embryonic compartments (*Haraguchi et al., 2001*). In zebrafish, quadruple mutants of *Mesp* genes (*Mesp-aa/-ab/-ba/-bb*) lack anterior somite segmentation while the positioning of posterior somite boundaries is unaffected, suggesting distinct requirements for *Mesp* genes in anterior and posterior somites (*Yabe et al., 2016*). Interestingly, during early ascidian development, *Mesp* is expressed in B7.5 founder cells that give rise to both CPM and anterior tail muscles (ATM) (*Satou et al., 2004*). In *Ciona*, the CPM precursors at the origin of heart and atrial siphon (pharyngeal) muscles depend on the ascidian homologs of *Mesp1*, *Islet1* and *Tbx1* (reviewed in [*Diogo et al., 2015*]), indicating that a conserved genetic network promotes chordate myogenesis in the anterior embryonic domain.

Our lineage analysis also reveals an unexpected contribution of *Islet1*-derived cells to the formation of cutaneous maximus and latissimus dorsi muscle progenitors (*Table 1*) (*Prunotto et al., 2004*; *Tremblay et al., 1998*). *Islet1* is activated in a subset of CPM progenitors giving rise to branchiomeric muscles and second heart field myocardium (*Cai et al., 2003*; *Harel et al., 2009*; *Nathan et al., 2008*). At the trunk level, while *Islet1* is widely expressed in the nervous system and in the LPM forming the hindlimb bud (*Cai et al., 2003*; *Yang et al., 2006*), to our knowledge its expression in somitic myogenic cells has not been reported. The cutaneous maximus and latissimus dorsi muscles are missing in both *Pax3* and *Met* mutants (*Prunotto et al., 2004*; *Tajbakhsh et al., 1997*; *Tremblay et al., 1998*). Therefore, the formation of the latissimus dorsi and cutaneous maximus muscles depends on a specific developmental program implicating *Pax3*, *Islet1* and *Met* genes. Given that the latissimus dorsi and cutaneous maximus participated in the gain in mobility of the forelimbs towards the shoulder girdle in tetrapods, our findings provide insights into their genetic and evolutionary origins.

Our detailed analysis of *Tbx1*- and *Pax3*-null mice on sections and in 3D reconstructions now provides a clarified view of neck muscle morphogenesis (*Table 2*). In both *Tbx1* and *Pax3* mutants, whereas the epaxial neck musculature is unaffected, the hypaxial muscles originating from anterior somites are perturbed with a more severe phenotype observed in *Tbx1* mutants (*Table 2*). Whereas no *Tbx1* expression has been reported in early myotomes in somites, *Tbx1* transcripts appear in hypaxial limb and tongue precursors after myogenic specification (*Grifone et al., 2008*; *Kelly et al., 2004*; *Zoupa et al., 2006*). *Tbx1*-null embryos show normal myotomal and limb muscle morphology while the hypoglossal cord is hypoplastic, resulting in reduced tongue musculature (*Table 2*) (*Grifone et al., 2008*; *Kelly et al., 2004*). Therefore, we cannot exclude the possibility that *Tbx1* is activated and plays a role after specification of neck hypaxial muscles (*Okano et al., 2008*;

*Zoupa et al., 2006*). The hypaxial muscle defects might also be secondary to a failure of caudal pharyngeal outgrowth (*Kelly et al., 2004*). While *Tbx1* acts cell autonomously in mesodermal progenitors (*Kong et al., 2014*; *Zhang et al., 2006*), its expression in pharyngeal endoderm might imply an indirect role in CPM myogenesis (*Arnold et al., 2006*). Defects in signaling from pharyngeal endoderm may explain the hypoglossal cord deficiency and the potential non-autonomous role for *Tbx1* in neck hypaxial myogenesis. Detailed analysis of muscle formation in conditional *Tbx1* mutants is needed to resolve the relative roles of *Tbx1* in neck myogenesis.

It has been shown that hypaxial muscles are perturbed to a greater extent than epaxial muscles in *Pax3* mutants (*Tajbakhsh et al., 1997*; *Tremblay et al., 1998*), suggesting a different requirement for *Pax3* in these muscle groups, possibly through differential gene regulation (*Brown et al., 2005*). An unexpected outcome of our analysis was that hypaxial neck muscles (derived from *Mesp1* and *Pax3* lineages) are less perturbed in *Pax3*-null mutants than hypaxial trunk/limb muscles (*Pax3* lineage only) that are severely altered or undetectable (*Table 2*). Our results indicate that *Pax3* is not essential for the formation of neck muscles derived from anterior somites in contrast to hypaxial muscles originating from more posterior somites. These observations support our model that a distinct genetic program governs somitic neck muscles compared to more posterior trunk muscles.

## Connectivity network of the neck and shoulders

Assessing the non-muscle contribution to the neck region is a major challenge due to the extensive participation of diverse cell types from different embryological origins. Previous studies in amphibians, chick and mouse reported that branchiomeric and hypobranchial connective tissue originates from NCCs (*Hanken and Gross, 2005*; *Köntges and Lumsden, 1996*; *Matsuoka et al., 2005*; *Noden, 1983*; *Olsson et al., 2001*; *Ziermann et al., 2018b*). It has been shown that the neural crest provides connective tissue for muscles that link the head and shoulders, whereas mesodermal cells give rise to connective tissue for muscles connecting the trunk and limbs (*Matsuoka et al., 2005*).

Our findings demonstrate that not all branchiomeric muscles are composed of neural crest-derived connective tissue, thereby redefining a new limit for NCC contribution to the neck and shoulders. Unexpectedly, we noted that the contribution of the neural crest lineage is limited in infrahyoid and posterior branchiomeric muscles that connect skeletal components of mesodermal origin. Indeed, it appears that the connective tissue of muscles that link exclusively mesodermal skeletal derivatives is of mesodermal origin. In contrast, the connective tissue of cucullaris-derived muscles is of a mixed origin, first developing in a cranial NCC domain at early stages, then expanding to incorporate connective tissue from both neural crest and LPM populations (*Figure 7B*). While NCCs are present in the anterior acromiotrapezius, sternocleidomastoid and infrahyoid muscles, contribution gradually decreases at posterior attachment sites and is undetectable at scapular level. In parallel, the LPM gives rise to shoulder skeletal components and to connective tissue at the attachment sites of associated musculature including trapezius muscles (*Figure 7C*). Therefore, the dual NCC/LPM origin of the trapezius connective tissue correlates with the embryonic origin of skeletal components to which it is connected.

*Wnt1^{Cre}* and *Sox10^{Cre}* NCC reporter mice were used to show that endochondral cells connecting the cucullaris-derived muscles on the scapula, clavicle and sternum share a common NCC origin with the connective tissue (*Matsuoka et al., 2005*). However, NCCs are not found in pectoral components of fish, axolotl and chick, while contribution to neurocranium is conserved, suggesting that NCC involvement in shoulder formation would be specific to mammals (*Epperlein et al., 2012*; *Kague et al., 2012*; *Piekarski et al., 2014*; *Ponomartsev et al., 2017*). In contrast to this view, our lineage analysis reveals that the neural crest lineage shows limited contribution to cucullaris connective tissue and does not form endochondral cells at the posterior attachment sites (*Figure 7C*). Differences in genetic lineage tracers and reagents might explain these discordant results (*Matsuoka et al., 2005*).

Taken together, our findings indicate that the gradient of neural crest and mesodermal contributions to neck connective tissue depends on the embryonic source of attachment sites. Therefore, it reveals that connective tissue composition in the neck region correlates with the cellular origin of associated skeletal components, independently of the myogenic source or ossification mode, forming a strong link between muscles and bones of the head, trunk and limb fields (*Figure 7D*).

## Evolutionary and clinical perspectives

Our findings demonstrate that the hybrid origin of the skeletal, connective tissue and muscle components of the neck is defined during early embryogenesis. The close proximity of neural crest, CPM, LPM and somitic populations is unique along the body plan and underscores the difficulty in defining their relative contributions to structures in the neck (*Figure 7A–B*). Our results refine the relative contributions of the neural crest and mesodermal derivatives in mouse, thereby providing a coherent view of embryonic components at the head-trunk interface in gnathostomes. Our study highlights the limited NCC contribution to posterior branchiomeric and infrahyoid muscle connective tissue, that is instead of mesodermal origin. This reinforces recent notions suggesting that the cranial NCCs and the postcranial rearrangement of mesodermal populations at the head-trunk interface had been central for the establishment of the neck during gnathostome evolution (*Adachi et al., 2018*; *Kuratani et al., 2018*; *Lours-Calet et al., 2014*; *Nagashima et al., 2016*; *Sefton et al., 2016*). The contribution of anterior mesoderm in the origin of the neck needs to be elucidated in future studies of gnathostomes.

Our study reveals that neck muscles develop in a complex domain that is distinct from the head and trunk (*Figure 7A–D*), and that might be a contributing factor to pathologies that affect subsets of neck muscles in specific myopathies (*Emery, 2002*; *Randolph and Pavlath, 2015*). In human, *TBX1* has been identified as a major candidate gene for 22q11.2 deletion syndrome (*Papangeli and Scambler, 2013*). Laryngeal malformations, esophageal dysmotility and shortened neck are frequent in patients. Moreover, the neck deficiencies might not be exclusively due to cervical spine abnormalities but also to neck muscle defects (*Hamidi et al., 2014*; *Leopold et al., 2012*; *Marom et al., 2012*). Therefore, our analysis of *Tbx1*-null mutants provides a better understanding of the etiology of the 22q11.2 deletion syndrome and has direct implications in establishing clinical diagnosis in cases where patients present failure in neck-associated functions.

# Materials and methods

**Key resources table**

| Reagent type (species) or resource | Designation | Source or reference | Identifiers | Additional information |
|---|---|---|---|---|
| Strain, strain background (*Mus musculus*) | B6D2F1/JRj | Janvier | | |
| Genetic reagent (*M. musculus*) | *Mef2c-AHF*[Cre] | PMID:16188249 | MGI:3639735 | Dr. Brian L Black (Cardiovascular Research Institute, University of California, USA) |
| Genetic reagent (*M. musculus*) | *Islet1*[Cre] | PMID:11299042 | MGI:2447758 | Dr. Thomas M Jessell (Howard Hughes Medical Institute, Columbia University, USA) |
| Genetic reagent (*M. musculus*) | *Mesp1*[Cre] | PMID:10393122 | MGI:2176467 | Pr. Yumiko Saga (National Institute of Genetics, Japan) |
| Genetic reagent (*M. musculus*) | *Pax3*[Cre] | PMID:22394517 | MGI:3573783 | Dr. Jonathan A. Epstein (Perelman Shool of Medicine, University of Pennsylvania, USA) |
| Genetic reagent (*M. musculus*) | *Myf5*[Cre] | PMID:17418413 | MGI:3710099 | Dr. Mario R Capecchi (Institute of Human Genetics, University of Utah, USA) |
| Genetic reagent (*M. musculus*) | *Wnt1*[Cre] | PMID:9843687 | MGI:J:69326 | Pr. Andrew P. McMahon (Keck School of Medicine of the University of Southern California, USA) |

*Continued on next page*

*Continued*

| Reagent type (species) or resource | Designation | Source or reference | Identifiers | Additional information |
|---|---|---|---|---|
| Genetic reagent (*M. musculus*) | *Prx1^{Cre}* | PMID:12112875 | MGI: J:77872 | Dr. Clifford J Tabin (Department of genetics, Harvard Medical School, USA) |
| Genetic reagent (*M. musculus*) | *Pax7^{GPL}* | PMID:19531352 | MGI:3850147 | Dr. Shahragim Tajbakhsh (Department of Developmental and Stem Cell Biology, Institut Pasteur, France) |
| Genetic reagent (*M. musculus*) | *Rosa26^{R-lacZ}* | PMID:9916792 | MGI:1861932 | Pr. Philippe Soriano (Icahn School of Medicine at Mt. Sinai, USA) |
| Genetic reagent (*M. musculus*) | *R26^{mTmG}* | PMID:17868096 | MGI:3716464 | Pr. Philippe Soriano (Icahn School of Medicine at Mt. Sinai, USA) |
| Genetic reagent (*M. musculus*) | *R26^{tdTomato}* | PMID:20023653 | MGI:3809524 | Dr. Hongkui Zeng (Allen Institute for Brain Science, USA) |
| Genetic reagent (*M. musculus*) | *Myf5^{nlacZ/+}* | PMID:8918877 | MGI:1857973 | Dr. Shahragim Tajbakhsh (Department of Developmental and Stem Cell Biology, Institut Pasteur, France) |
| Genetic reagent (*M. musculus*) | *Tbx1-null* | PMID:11242110 | MGI:2179190 | Dr. Virginia Papaioannou (Department of Genetics and Development, Columbia University Medical Center, USA) |
| Antibody | Chicken polyclonal anti-β-gal | Abcam | Cat. #: ab9361 | IF (1:1000) |
| Antibody | Rabbit polyclonal anti-β-gal | MP Biomedicals | Cat. #: MP 559761 | IF (1:750) |
| Antibody | Chicken polyclonal anti-GFP | Aves Labs | Cat. #: 1020 | IF (1:500) |
| Antibody | Chicken polyclonal anti-GFP | Abcam | Cat. #: 13970 | IF (1:1000) |
| Antibody | Mouse monoclonal IgG1 anti-Islet1 | DSHB | Cat. #: 40.2D6 | IF (1:1000) |
| Antibody | Mouse monoclonal IgG1 anti-My32 | Sigma | Cat. #: M4276 | IF (1:400) |
| Antibody | Mouse monoclonal IgG1 anti-Myod | Dako | Cat. #: M3512 | IF (1:100) |
| Antibody | Mouse monoclonal IgG1 anti-Pax7 | DSHB | Cat. #: AB_528428 | IF (1:20) |
| Antibody | Rabbit polyclonal anti-Tcf4 | Cell Signalling | Cat. #: C48H11 | IF (1:150) |
| Antibody | Mouse monoclonal IgG1 anti-Tnnt3 | Sigma | Cat. #: T6277 | IF (1:200) |
| Antibody | Rabbit polyclonal anti-Tomato | Clontech | Cat. #: 632496 | IF (1:500) |
| Antibody | Mouse monoclonal IgG2a anti-Pax7 | Ozyme | Cat. #: BLE801202 | IF (1:1000) |
| Software, algorithm | GE phoenix datos\|x 2.0 | GE Sensing and Inspection Technologies GmbH | | |
| Software, algorithm | 3D PDF maker | SolidWorks Corporation | | |
| Software, algorithm | Zen | Zeiss | | |

*Continued on next page*

*Continued*

| Reagent type (species) or resource | Designation | Source or reference | Identifiers | Additional information |
|---|---|---|---|---|
| Chemical compound, drug | X-gal | Fisher | Cat. #: 10554973 | |
| Chemical compound, drug | paraformaldehyde | Electron Microscopy Sciences | Cat. #: 15710 | |
| Chemical compound, drug | Triton X-100 | Sigma | Cat. #: T8787 | |
| Chemical compound, drug | Tween 20 | Sigma | Cat. #: P1379 | |
| Chemical compound, drug | Histoclear II | National Diagnostics | Cat. #: HS-202 | |

## Animals

Animals were handled as per European Community guidelines and the ethics committee of the Institut Pasteur (CTEA) approved protocols (APAFIS#6354–20160809 l2028839). Males carrying the *Cre* driver gene, *Mef2c-AHF$^{Cre}$* (**Verzi et al., 2005**), *Islet1$^{Cre}$* (**Srinivas et al., 2001**), *Mesp1$^{Cre}$* (**Saga et al., 1999**), *Pax3$^{Cre}$* (**Engleka et al., 2005**), *Myf5$^{Cre}$* (**Haldar et al., 2007**), *Wnt1$^{Cre}$* (**Danielian et al., 1998**), *Prx1$^{Cre}$* (**Logan et al., 2002**), were crossed to reporter females from previously described lines including *Pax7$^{GPL}$* (**Sambasivan et al., 2013**), *Rosa26$^{R-lacZ}$* (*R26R*) (**Soriano, 1999**), *R26$^{mTmG}$* (**Muzumdar et al., 2007**) and *R26$^{tdTomato}$* (**Madisen et al., 2010**). *Myf5$^{nlacZ/+}$* KI mice and mice carrying the *Tbx1$^{tm1pa}$* allele (referred to as *Tbx1*-null) were previously described (**Jerome and Papaioannou, 2001**; **Kelly et al., 2004**; **Tajbakhsh et al., 1996**). To generate experimental *Pax3*-null fetuses, *Pax3$^{WT/Cre}$* males and females were intercrossed (**Engleka et al., 2005**) (n = 5 *Tbx1* and *Pax3* mutants analysed including n = 2 by µCT scanning). Mice were crossed and maintained on a B6D2F1/JRj background and genotyped by PCR. Mouse embryos and fetuses were collected between E9.5 and E18.5, with noon on the day of the vaginal plug considered as E0.5.

## X-gal and immunofluorescence stainings

Whole-mount samples were analysed for beta-galactosidase activity with X-gal (0.6 mg/ml) in 1X PBS buffer (D1408, Sigma, St. Louis, MO) containing 4 mM potassium ferricyanide, 4 mM potassium ferrocyanide, 0.02% NP-40 and 2 mM $MgCl_2$ as previously described (**Comai et al., 2014**). For immunostaining on cryosections, foetuses were fixed 3 hr in 4% paraformaldehyde (PFA) (15710, Electron Microscopy Sciences, Hatfield, PA) 0.5% Triton X-100 (T8787, Sigma) at 4°C, washed overnight at 4°C in PBS 0.1% Tween 20 (P1379, Sigma), cryopreserved in 30% sucrose in PBS and embedded in OCT for 12–16 µm sectioning with a Leica cryostat (CM3050 S, Leica, Wetzlar, Germany). Cryosections were dried for 30 min and washed in PBS. For immunostaining on paraffin sections, samples were fixed overnight in 4% PFA, dehydrated in graded ethanol series and penetrated with Histoclear II (HS-202, National Diagnostics, Atlanta, GA), embedded in paraffin and oriented in blocks. Paraffin blocks were sectioned at 10–12 µm using a Leica microtome (Reichert-Jung 2035). Sections were then deparaffinized and rehydrated by successive immersions in Histoclear, ethanol and PBS. Samples were then subjected to antigen retrieval with 10 mM Citrate buffer (pH 6.0) using a 2100 Retriever (Aptum Biologics, Rownhams, UK).

Rehydrated sections were blocked for 1 hr in 10% normal goat serum, 3% BSA, 0.5% Triton X-100 in PBS. Primary antibodies were diluted in blocking solution and incubated overnight at 4°C. Primary antibodies included the following: β-gal (1/1000, chicken polyclonal, ab9361, Abcam, Cambridge, UK; 1/750, rabbit polyclonal, MP 559761, MP Biomedicals, Illkirch, France), GFP (1/500, chick polyclonal, 1020, Aves Labs, Tigard, OR; 1/1000, chick polyclonal, 13970, Abcam), Islet1 (1/1000, mouse monoclonal IgG1, 40.2D6, DSHB), My32 (1/400, mouse monoclonal IgG1, M4276, Sigma), Myod (1/100, mouse monoclonal IgG1, M3512, Dako, Santa Clara, CA), Pax7 (1/20, mouse monoclonal IgG1, AB_528428), Tcf4 (1/150, rabbit polyclonal, C48H11, Cell Signalling, Leiden, Netherlands), Tnnt3 (1/200, monoclonal mouse IgG1, T6277, Sigma), Tomato (1/500, rabbit polyclonal, 632496, Clontech, Shiga, Japan; 1/250, chick polyclonal, 600-901-379, Rockland, Pottstown, PA) and Tuj1 (1/1000, monoclonal mouse IgG2a, BLE801202, Ozyme, Montigny-le-Bretonneux, France). After 3 rounds of

15 min washes in PBS 0.1% Tween 20, secondary antibodies were incubated in blocking solution 2 hr at RT together with 1 µg/ml Hoechst 33342 to visualize nuclei. Secondary antibodies consisted of Alexa 488, 555 or 633 goat anti-rabbit, anti-chicken or anti-mouse isotype specific (1/500, Jackson Immunoresearch, Cambridgeshire, UK). After 3 rounds of 15 min washes in PBS 0.1% Tween 20, slides were mounted in 70% glycerol for analysis.

For whole-mount immunofluorescence staining, embryos were dissected in PBS, fixed in 4% PFA, washed in PBS and stored at −20°C in 100% methanol. After rehydration in PBS, whole mount immunostainings were performed incubating the primary and secondary antibodies for 3 days each. Samples were cleared using benzyl alcohol/benzyl benzoate (BABB) clarification method (*Yokomizo et al., 2012*).

### µCT scan analysis

For µCT scan analysis, the fetuses were treated with the phosphotungstic acid (PTA) contrast agent to well reveal skeletal and muscle structures. After dissection of the cervical region (including the mandible and scapular components, see *Figure 2—figure supplement 1*), the fetuses were fixed in 4% PFA for 24 hr at 4°C. Samples were then additionally fixed and dehydrated by exchanging the fixative and washing solutions to incrementally increasing ethanol concentrations (30%, 50%, 70%) with 2 days in each concentration to minimize the shrinkage of tissues. To start the contrasting procedure, the embryos were firstly incubated in ethanol-methanol-water mixture (4:4:3) for 1 hr and then transferred for 1 hr into 80% and 90% methanol solution. The staining procedure was then performed for 10 days in 90% methanol 1.5% PTA solution (changed every day with fresh solution) to ensure optimal penetration of the contrast agent. Staining was followed by rehydration of the samples in methanol-grade series (90%, 80%, 70%, 50% and 30%) and stored in sterile distilled water. The samples were placed in polypropylene tubes and embedded in 1% agarose gel to avoid movement artefacts during measurements. µCT scanning was performed using laboratory system GE Phoenix v|tome|x L 240 (GE Sensing and Inspection Technologies GmbH, Hamburg, Germany), equipped with a nanofocus X-ray tube with maximum power of 180 kV/15 W and a flat panel detector DXR250 with 2048 × 2048 pixel2, 200 × 200 µm$^2$ pixel size. The µCT scan was carried out at 60 kV acceleration voltage and 200 µA tube current with voxel size of 5.7 µm for all samples. The beam was filtered by a 0.2 mm aluminium filter. The 2200 projections were taken over 360° with exposure time of 900 ms. The tomographic reconstructions were done using the software GE phoenix datos|x 2.0 (GE Sensing and Inspection Technologies GmbH) and data segmentations and visualizations were performed by combination of software VG Studio MAX 2.2 (Volume GraphicsGmbH, Heidelberg, Germany) and Avizo 7.1 (Thermo Fisher Scientific, Waltham, MA) according to (*Tesařová et al., 2016*). The interactive 3D PDFs were set up using 3D PDF maker software.

### Imaging

Images were acquired using the following systems: a Zeiss Axio-plan equipped with an Apotome, a Zeiss stereo zoom microscope V16 or a Zeiss LSM 700 laser-scanning confocal microscope with ZEN software (Carl Zeiss, Oberkochen, Germany). For whole-mount rendering, acquired Z-stacks were 3D reconstructed using Imaris software. All images were assembled in Adobe Photoshop (Adobe Systems, San Jose, CA).

## Acknowledgements

We thank Drs. Claudio Cortes and Françoise Helmbacher for providing transgenic mice and Mirialys Gallardo for technical assistance. We also thank Cécile Mazur for illustrations.

## Additional information

### Funding

| Funder | Grant reference number | Author |
| --- | --- | --- |
| Institut Pasteur | | Eglantine Heude<br>Alexandre Grimaldi<br>Shahragim Tajbakhsh |

| Agence Nationale de la Recherche | | Eglantine Heude<br>Alexandre Grimaldi<br>Shahragim Tajbakhsh |
|---|---|---|
| Centre National de la Recherche Scientifique | | Eglantine Heude<br>Estelle Jullian<br>Noritaka Adachi<br>Alexandre Grimaldi<br>Robert G Kelly<br>Shahragim Tajbakhsh |
| French Muscular Dystrophy Association | | Eglantine Heude<br>Alexandre Grimaldi<br>Robert G Kelly<br>Shahragim Tajbakhsh |
| Central European Institute of Technology | | Marketa Tesarova<br>Tomas Zikmund<br>Jozef Kaiser |
| March of Dimes Foundation | | Elizabeth M Sefton<br>Gabrielle Kardon |
| National Institutes of Health | | Elizabeth M Sefton<br>Gabrielle Kardon |
| Fondation pour la Recherche Médicale | | Estelle Jullian<br>Noritaka Adachi<br>Robert G Kelly |
| Fondation Leducq | | Estelle Jullian<br>Noritaka Adachi<br>Robert G Kelly |
| Yamada Science Foundation | | Noritaka Adachi<br>Robert G Kelly |
| Bourses du Gouvernement Français | | Noritaka Adachi |
| project CEITEC 2020 | | Marketa Tesarova<br>Tomas Zikmund<br>Jozef Kaiser |
| The Ministry of Education, Youth and Sports of the Czech Republic under the National Sustainability Programme II | | Marketa Tesarova<br>Tomas Zikmund<br>Jozef Kaiser |
| CEITEC Nano Research Infrastructure | MEYS CR, 2016–2019 | Marketa Tesarova<br>Tomas Zikmund<br>Jozef Kaiser |

The funders had no role in study design, data collection and interpretation, or the decision to submit the work for publication.

## Author contributions

Eglantine Heude, Conceptualization, Validation, Investigation, Visualization, Methodology, Writing—original draft, Writing—review and editing; Marketa Tesarova, Resources, Data curation, Formal analysis, Investigation, Visualization, Methodology; Elizabeth M Sefton, Validation, Investigation, Writing—review and editing; Estelle Jullian, Validation, Investigation; Noritaka Adachi, Alexandre Grimaldi, Investigation, Writing—review and editing; Tomas Zikmund, Resources, Supervision; Jozef Kaiser, Resources, Supervision, Funding acquisition, Project administration; Gabrielle Kardon, Robert G Kelly, Resources, Supervision, Funding acquisition, Project administration, Writing—review and editing; Shahragim Tajbakhsh, Conceptualization, Resources, Supervision, Funding acquisition, Validation, Visualization, Methodology, Project administration, Writing—review and editing

## Author ORCIDs

Elizabeth M Sefton http://orcid.org/0000-0001-6481-612X
Noritaka Adachi http://orcid.org/0000-0002-9482-8436
Shahragim Tajbakhsh http://orcid.org/0000-0003-1809-7202

## Ethics

Animal experimentation: Animals were handled as per European Community guidelines and the ethics committee of the Institut Pasteur (CTEA) approved protocols. (APAFIS#6354-20160809I2028839)

## Decision letter and Author response

Decision letter https://doi.org/10.7554/eLife.40179.028
Author response https://doi.org/10.7554/eLife.40179.029

---

# Additional files

## Supplementary files

• Supplementary file 1. Interactive 3D neck reconstruction of a E18.5 control fetus. Download PDF for full details.
DOI: https://doi.org/10.7554/eLife.40179.023

• Supplementary file 2. Interactive 3D neck reconstruction of a E18.5 *Tbx1*-null fetus. Download PDF for full details.
DOI: https://doi.org/10.7554/eLife.40179.024

• Supplementary file 3. Interactive 3D neck reconstruction of a E18.5 *Pax3*-null fetus. Download PDF for full details.
DOI: https://doi.org/10.7554/eLife.40179.025

• Transparent reporting form
DOI: https://doi.org/10.7554/eLife.40179.026

## Data availability

All data generated or analysed during this study are included in the manuscript and supporting files.

---

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
