## [Decision Letter]

Thank you for submitting your article "Unique morphogenetic signatures define mammalian neck muscles and associated connective tissues" for consideration by *eLife*. Your article has been reviewed by two peer reviewers, and the evaluation has been overseen by a Reviewing Editor and Didier Stainier as the Senior Editor. The following individual involved in review of your submission has agreed to reveal his identity: Robert Knight (Reviewer #2).

The reviewers have discussed the reviews with one another and the Reviewing Editor has drafted this decision to help you prepare a revised submission.

Summary:

This manuscript examines the complex lineage and genetic regulation of the muscles in the lower head and neck. To characterize lineage, Heude et al. use mouse lines expressing Cre under the control of relatively well-characterised regulatory elements expressed in subsets of the musculature (*Islet1, Mesp1, Pax3*), the neural crest (*Wnt1*) and trunk lateral mesoderm (*Prx1*). They also examine the phenotype of these head and neck muscles in mice lacking *Tbx1* and *Pax3*. They focus on the cucullaris muscles that connect head and neck skeletal elements. This is interesting from an evolutionary perspective as an essential adaptation of gnathostomes allowing movement of the head relative to the body, and also because certain genetic syndromes differentially affect muscles in the head and neck. The origin of this muscle group has so far been controversial. The main findings of this work are that differential expression of *Mesp1, Islet1* and *Pax3* distinguish the head, neck and shoulder muscles, and that the cucullaris originates from branchiomeric (head) paraxial mesoderm, not trunk lateral mesoderm as previously suggested. *Tbx1* is essential for subsets of head-derived muscle while *Pax3* is required for anterior somite-derived muscle.

This is an excellent study of some complex developmental anatomy, presented in a generally beautifully clear and informative way. The results are significant from an evolutionary perspective and relevant to mechanisms that may underlie various musculoskeletal abnormalities in humans. The three-dimensional micro-CT scans presented as interactive PDFs are particularly informative and an ideal way to display complex data of this type – the sections can display only a small fraction of this information.

Essential revisions:

The authors are requested to clarify the following issues:

1) Figure 7A displays a hypothetical early origin of the head and neck muscles that is not presented in the data. It crucially shows bilateral 'fingers' of cranial paraxial mesoderm sitting immediately lateral to somites 1-3, with lateral mesoderm situated more lateral to this. Is there any reason why there are no stained embryos of this stage (or even at E9.5) in the series presented in Figure 1? Without this, it could be argued that the controversial question of paraxial versus lateral plate mesoderm origin of the cucullaris muscle has not been completely resolved. For example *Prx1* may not be expressed in a small subset of lateral plate mesoderm corresponding to future cucullaris, or conversely *Mesp1/Islet1* may secondarily be expressed in a cucullaris anlage that shares a lineage origin with trunk lateral plate mesoderm. If the possibility of a lateral mesoderm origin of cucullaris muscle cannot be ruled out by the current approach, this should be acknowledged in the text.

2) Please indicate how expression of the constitutive reporters of *Mesp1*, a subset of *Mef2c, Pax3* and *Islet1* expression is affected in the *Tbx1* and *Pax3* mutant mice that are described later in the paper, as this could allow expression of the reporters to be correlated with muscle phenotypes and so make the association between expression and phenotypic outcome clearer. Please also comment on whether the expression of Islet1 in somitic myogenic cells is dependent on *Tbx1* or *Pax3* during early stages of muscle formation.

3) The use of *Pax3* transgenic line for tracking somitic mesoderm may complicate analysis as it is also expressed in CNCC. The authors note this in the third paragraph of the subsection “Distinct myogenic programs define neck muscle morphogenesis” as they observe a contribution of *Pax3*-expressing cells to the cucullaris but do not address this point further. As a central tenet of the paper is that there is no somite-derived contribution to the cucullaris this should be addressed by comparing *Pax3* reporter expression to NCC markers, since an alternative interpretation is that there is a contribution of somite-derived cells to this muscle that migrate to the anlage after E10.5.

---

## [Author Response]

Essential revisions:The authors are requested to clarify the following issues:1) Figure 7A displays a hypothetical early origin of the head and neck muscles that is not presented in the data. It crucially shows bilateral 'fingers' of cranial paraxial mesoderm sitting immediately lateral to somites 1-3, with lateral mesoderm situated more lateral to this. Is there any reason why there are no stained embryos of this stage (or even at E9.5) in the series presented in Figure 1? Without this, it could be argued that the controversial question of paraxial versus lateral plate mesoderm origin of the cucullaris muscle has not been completely resolved. For example Prx1 may not be expressed in a small subset of lateral plate mesoderm corresponding to future cucullaris, or conversely Mesp1/Islet1 may secondarily be expressed in a cucullaris anlage that shares a lineage origin with trunk lateral plate mesoderm. If the possibility of a lateral mesoderm origin of cucullaris muscle cannot be ruled out by the current approach, this should be acknowledged in the text.

The hypothetical early origin of postcranial CPM extension has been proposed on the basis of our results and data reported previously. Unfortunately, there are no specific markers or reporters available to date to trace the early CPM presented in Figure 7A. Our genetic lineage analysis at later stages, in combination with previous work involving innervation studies, grafting experiments, and retrospective clonal analyses indicate that the mesoderm adjacent to the 3 first somites correspond to a posterior branchiomeric population in the early tetrapod embryo (see also Couly et al., Development, 1992). The analysis of the *Prx1* lineage, that appears to define the trunk LPM field, corroborates that the cucullaris muscles are clonally related to the posterior branchiomeric muscles as shown previously through retrospective clonal analysis (Lescroart et al., 2015). We have modified the Discussion to clarify this point (subsection “Branchiomeric origin of cucullaris-derived muscles”, first paragraph). Moreover, *Mesp1* does not seem to be secondarily expressed in the cucullaris anlagen. We propose that *Mesp1* activation in the nascent mesoderm during gastrulation is more likely defining both CPM and anterior somites.

2) Please indicate how expression of the constitutive reporters of Mesp1, a subset of Mef2c, Pax3 and Islet1 expression is affected in the Tbx1 and Pax3 mutant mice that are described later in the paper, as this could allow expression of the reporters to be correlated with muscle phenotypes and so make the association between expression and phenotypic outcome clearer. Please also comment on whether the expression of Islet1 in somitic myogenic cells is dependent on Tbx1 or Pax3 during early stages of muscle formation.

Expression of the constitutive reporters in *Tbx1* and *Pax3* mutants has not been analysed in the present study. However, based on lineage analyses and anatomical descriptions, we have defined the genetic embryological origin of the different head/neck/pectoral muscles analysed in mutant foetuses (Figures 5-6, see also our new Figure 7—figure supplement 1). Please find the revised Table 2 that now highlights the myogenic lineages used for the development of the muscle groups, correlating lineage analysis and phenotypic outcomes. Please refer to the colour codes used for the different myogenic programs reported in the main Figures (1, 2, 3, 6, 7), in Figure 7—figure supplement 1 and in the 3D interactive PDFs (Supplementary files 1-3).

To address *Islet1* contribution to the somitic mesoderm, we have modified Figure 1—figure supplement 2C. The results now show that *Islet1* expression is observed in *Pax3*-derived cells in the precursor of the cutaneous maximus and latissimus dorsi somitic muscles (*clp*), completing information reported in Table 1 (footnote **). We have now modified the Results section accordingly (subsection “Distinct myogenic programs define neck muscle morphogenesis”, third paragraph). Moreover, we describe in the Discussion that these muscles depend on a specific myogenic program implicating *Pax3* and *Islet1* genes and that they are absent in *Pax3* mutants (subsection “A unique genetic program for somite-derived neck muscles”, third paragraph). In contrast, *Tbx1* expression has never been reported during early somitic muscle development, indicating that *clp* formation is dependent on *Islet1* and *Pax3* but not on *Tbx1*.

3) The use of Pax3 transgenic line for tracking somitic mesoderm may complicate analysis as it is also expressed in CNCC. The authors note this in the third paragraph of the subsection “Distinct myogenic programs define neck muscle morphogenesis” as they observe a contribution of Pax3-expressing cells to the cucullaris but do not address this point further. As a central tenet of the paper is that there is no somite-derived contribution to the cucullaris this should be addressed by comparing Pax3 reporter expression to NCC markers, since an alternative interpretation is that there is a contribution of somite-derived cells to this muscle that migrate to the anlage after E10.5.

To address this point, we have added an analysis of the *Pax3* and *Mesp1* contributions to the perinatal trapezius at high magnification (Figure 3—figure supplement 4C-D). This new result demonstrates that the *Pax3*-derived cells in the trapezius correspond to the Tcf4-positive fibroblast population associated with myofibres, similar to that observed in the cranial masseter and tongue muscles, that are composed of NCC-derived connective tissue (Figure 3—figure supplement 2A-B’, Figure 3—figure supplement 4A-B, G). We have modified the Results section accordingly (subsection “Dual neural crest and mesodermal origins of neck connective tissues”, third paragraph). Taken together, our results exclude the possibility of a *Pax3* lineage contribution to cucullaris myofibres.